# SAM68 directs STING signaling to apoptosis in macrophages
Demi van der Horst[1], Naziia Kurmasheva[1,4], Mikkel H. S. Marqvorsen[1,4], Sonia Assil[1,4], Anna H. F. Rahimic [1], Christoph F. Kollmann [1], Leandro Silva da Costa[1], Qi Wu [1], Jian Zhao[1], Eleonora Cesari[2], Marie B. Iversen[1], Fanghui Ren[1], Trine I. Jensen [1], Ryo Narita[1], Vivien R. Schack[1], Bao-cun Zhang [1], Rasmus O. Bak [1], Claudio Sette[2,3], Robert A. Fenton[1], Jacob G. Mikkelsen[1], Søren R. Paludan [1,5] ✉ & David Olagnier [1,5] ✉

DNA is a danger signal sensed by cGAS to engage signaling through STING to activate innate immune functions. The best-studied downstream responses to STING activation include expression of type I interferon and inflammatory genes, but STING also activates other pathways, including apoptosis. Here, we report that STING-dependent induction of apoptosis in macrophages occurs through the intrinsic mitochondrial pathway and is mediated via IRF3 but acts independently of gene transcription. By intersecting four mass spectrometry datasets, we identify SAM68 as crucial for the induction of apoptosis downstream of STING activation. SAM68 is essential for the full activation of apoptosis. Still, it is not required for STING-mediated activation of IFN expression or activation of NF-κB. Mechanistic studies reveal that protein trafficking is required and involves SAM68 recruitment to STING upon activation, with the two proteins associating at the Golgi or a post-Golgi compartment. Collectively, our work identifies SAM68 as a STING-interacting protein enabling induction of apoptosis through this DNA-activated innate immune pathway.

The innate immune system represents the first line of host defense and utilizes pattern recognition receptors (PRR)s to sense conserved or mis-located molecules, thus initiating immune responses[1]. Over the years, sensing pathogens and tumors through detecting their nucleic acids has emerged as an important mechanism for host protection by the innate immune system[1]. For instance, DNA was discovered more than 50 years ago to be highly immuno-stimulatory[2,3], and we now know that the major DNA sensors are Toll-like receptor 9 (TLR9) located in the endosomes, and the cytosolic DNA sensors absent in melanoma 2 (AIM2) and cyclic GMP-AMP (cGAMP) synthase (cGAS)[4–6].

One of the main characteristics of the cytosolic DNA-driven immune response is the induction of antiviral cytokines, most notably type I interferons (IFNs)[7]. This occurs via cGAS and the endoplasmic reticulum-localized adapter protein stimulator of interferon (IFN) genes (STING) relaying to signaling to the antiviral type I IFN response[6,8–10]. Upon recognition of DNA, the cGAS enzyme is activated to catalyze the formation of the cyclic dinucleotide (CDN) 2'3' cGAMP', which binds to STING[8] to promote its trafficking to the Golgi, where TANK-binding kinase 1 (TBK1) is recruited[11]. Subsequently, TBK1 phosphorylates STING and the transcription factor IRF3, thus promoting transcription of type I IFN genes[12]. In addition, cGAS-STING signaling can activate the transcription factor NF-kB, and the cytosolic catabolic process of autophagy[13–16]. Interestingly, studies have highlighted the IFN-independent activities of STING in controlling virus infection, tumor immune evasion, and adaptive immunity[17,18]. Termination of STING signaling includes multiple mechanisms, including autophagy of STING molecules and sorting of phosphorylated STING molecules into clathrin-coated transport vesicles for lysosomal degradation[19,20]. STING signaling was also recently reported to be interrupted through STING degradation via endosomal sorting complexes required for transport (ESCRT)-mediated microautophagy[21,22].

Programmed cell death (PCD) is a term for cellular modalities of cell death occurring through molecularly defined and specific signaling pathways. Critical forms of PCD are apoptosis, pyroptosis, and necroptosis[23]. In addition, there are also other less well-characterized forms of PDC,

[1]Department of Biomedicine, Aarhus University, Høegh Guldbergsgade 10, 8000 Aarhus C, Denmark. [2]GSTEP-Organoids Core Facility, IRCCS Fondazione Policlinico Agostino Gemelli, 00168 Rome, Italy. [3]Department of Neuroscience, Section of Human Anatomy, Catholic University of the Sacred Hearth, 00168 Rome, Italy. [4]These authors contributed equally: Naziia Kurmasheva, Mikkel H. S. Marqvorsen, Sonia Assil. [5]These authors jointly supervised this work: Søren R. Paludan, David Olagnier. ✉e-mail: srp@biomed.au.dk; olagnier@biomed.au.dk

including autophagic cell death and lysosomal cell death[24,25]. Pyroptosis and necroptosis represent two forms of necrotic PCD that stimulate inflammation. They are executed upon forming Gasdermin and mixed-lineage kinase domain-like pores in the plasma membrane downstream of signaling by inflammasomes and RIPK1/3 complexes, respectively[23]. Apoptosis is a non-inflammatory form of PCD, engaging caspase 3 and 7, which cleave specific substrates to induce DNA fragmentation, cytoskeleton proteolysis, and laminin degradation, eventually causing cell shrinkage and cell death without leakage of cytosolic content[26]. Caspase 3/7 activation can be achieved from both extracellular and intracellular stimuli. The intrinsic apoptosis pathway is activated upon leakage of cytochrome C from the mitochondria through the voltage-dependent anion-channel and pores formed by oligomers of the protein BAX and BAK[26]. Cytosolic cytochrome C enables activation of the apoptosome complex, which initiates the caspase activation cascades ending with apoptosis.

Subsequent to the discovery of cGAS-STING signaling inducing immuno-modulatory gene expression, numerous studies have demonstrated that cGAS-STING signaling also leads to PCD. This includes apoptosis, necroptosis, and lysosomal cell death[25,27–31], with apoptosis being particularly well documented[32]. At the physiological level, STING-induced apoptosis was reported to promote cancer cell killing in a mouse model for T-cell lymphoma cells[28], but also that tumors induce STING-mediated T cell death to evade immune control[18]. This may also have implications for cancer immunotherapy, where STING activation is being tested for its potential to promote anti-tumor T cell responses. In addition, we recently reported that cGAS-STING-dependent apoptosis in brain immune cells exerts negative regulation of immune activation during herpes simplex virus (HSV)-1 infection[29]. Finally, patients with constitutive STING activation due to gain-of-function mutations exhibit lymphopenia and accelerated cell death of T cells and monocytes in vitro[33]. Thus, STING-induced apoptosis is of physiological and therapeutic importance, but the underlying mechanism remains largely unresolved. One of the first studies to identify STING-dependent apoptosis reported formation of a complex between BAX and IRF3 in primary human monocytes, inducing apoptosis through the intrinsic pathway[31]. These findings shared mechanistic overlap with what had been reported in earlier work by Chattopadhyay et al who demonstrated that IRF3 could activate and bind BAX at the mitochondria to elicit apoptosis in response to cytoplasmic dsRNA[34]. Other studies also suggested the involvement of a STING-driven transcriptional program controlling apoptosis[28,29]. Here we report that STING interacts with SAM68 upon activation in macrophages, and this enables direct signaling to the IRF3-dependent mitochondrial apoptosis pathway, independent of de novo protein synthesis and IFNAR signaling. Depletion of SAM68 attenuates STING-induced apoptosis but not the induction of IFN and inflammatory gene expression. This suggests that SAM68 specifically directs STING signaling towards apoptosis.

## Results

### STING induces cell death through the mitochondrial apoptosis pathway in macrophages

STING has been reported to promote PCD through several pathways[24,25,28,30,31,35,36], but information on the mechanisms remains sparse. Stimulation of STING using exogenous or lipofectamine-delivered cGAMP (Supplementary Fig. S1a), or engagement of STING with the synthetically modified agonist 2'3-cGAM(PS)$_2$ or the non-cyclic dinucleotide diABZI led to the cleavage of the apoptotic markers Caspase 3 and PARP in PMA-differentiated THP1 macrophage-like cells (PMA-THP1) (Supplementary Fig. S1b). In PMA-THP1 and primary human monocyte-derived macrophages (hMDM), we observed that dsDNA induced cell death and this was inhibited by the pan-caspase inhibitor Z-VAD and the caspase 3 inhibitor Z-DEVD, but not by inhibitors of pyroptosis and necroptosis pathways (Fig. 1a, b, Supplementary Fig. S2a). The observed cleavage of Caspase 3 and PARP was blocked by Z-VAD (Fig. 1c). Moreover, treatment of cells with cGAMP or dsDNA induced annexin V and Propidium Iodide (PI) positive staining, with a strong correlation between the two signals at the single cell

level (Fig. 1d–f). To exclude the possibility that pyroptosis was involved in cGAMP or dsDNA-induced cell death, the same experiments were performed in ASC (apoptosis-associated speck-like protein containing a CARD)-lacking cells, a critical adapter protein driving inflammasome activation and pyroptosis. Cell death induction was not affected by ASC-deficiency (Supplementary Fig. S2b–e), thus suggesting apoptotic cell death and unequivocally excluding pyroptosis. To discriminate between the extrinsic and the intrinsic apoptosis pathways, we double-stained cells for loss of mitochondrial potential using a MitoTracker probe and annexin V (early apoptosis). We observed both to be elevated in cGAMP- or dsDNA-treated cells with strong correlation between the two signals at the single cell level (Fig. 1g–i). In line with this, cGAMP-treatment led to the release of cytochrome C into the cytoplasm (Fig. 1j). To unequivocally confirm that cGAMP induced cell death through the intrinsic pathway, expression of both BAX and BAK1 were depleted by CRISPR/Cas9 genome editing. Simultaneous targeting of BAX and BAK1 led to a reduction in STING-driven LDH release following cGAMP stimulation and accumulation of the apoptosis markers cleaved Caspase 3 and PARP (Fig. 1k, l). To functionally validate the BAX/BAK1 double KO in PMA-THP1, we used a combination of BH3 mimetic ABT-737 and MCL1 inhibitor S-63845 as a positive control for intrinsic apoptosis (Supplementary Fig. S2f, g).

Overall, these data demonstrate that STING-induced PCD in THP1 occurs through the intrinsic mitochondrial apoptosis pathway in human THP1-differentiated macrophages.

### STING induces apoptosis dependent on IRF3 but not de novo protein synthesis

STING-dependent apoptosis has been reported to differ significantly in different cell types concerning the dependence on TBK1-IRF3 and the dependence on de novo protein synthesis[28,29,31,35,37]. To characterize the signaling requirements in THP1-derived macrophages, we treated a panel of genome-edited cells with cGAMP and DNA and evaluated the apoptotic response. These data showed that cells lacking STING, TBK1, or IRF3 could not induce apoptosis (Fig. 2a–c; Supplementary Fig. S2h, i). Interestingly, however, induction of apoptosis was not compromised in IFNAR2-deficient cells (Fig. 2d, e) or in cells where translation has been blocked by cyclo-heximide treatment (Fig. 2f). To examine the molecular requirements for STING to support apoptosis, we reconstituted STING-deficient cells with a series of mutants and examined for cGAMP-activated apoptosis. We observed an absolute requirement for the C-terminal tail and the residue S366 of STING for IRF3 recruitment (Fig. 2g). Finally, 6 h after STING engagement, IRF3 was shown to be localized in both cellular compartments: nuclei and mitochondria, while it was distributed throughout the cytoplasm at a steady state and predominantly in the nucleus 3 h post cGAMP stimulation (Fig. 2h, i and Supplementary Fig. S3). Collectively, these data demonstrate that STING-induced apoptosis in macrophages depends on IRF3 without the need for de novo protein synthesis.

### SAM68/KHDRBS1 is essential for the induction of apoptosis downstream of STING in macrophages

AIFM1 was recently identified as involved in STING-dependent apoptosis in T lymphocytes[38]. However, depletion of AIFM1 expression did not affect cGAMP-induced apoptosis in PMA THP1 macrophages (Supplementary Fig. S4a). To identify novel STING interacting factors that may be involved in the induction of apoptosis, we performed co-immunoprecipitation of STING in the presence or absence of cGAS. We uncovered novel interacting proteins via mass spectrometry (Supplementary Data 1). We combined this dataset with three other mass spectrometry datasets, namely cGAS-interacting proteins[39], DNA-activated phospho-proteome[40], and dsDNA interactome[41] (Fig. 3a). This led to the identification of 8 proteins shared between at least 3 of the datasets (Fig. 3b). To explore whether any of these proteins were involved in STING-driven cell death, we treated THP1 cells with specific gRNA/Cas9 RNP complexes (two different sequence target per gene). We examined dsDNA-induced immune and apoptotic responses (Supplementary Fig. S4b). Among the genes targeted, only gRNAs specific

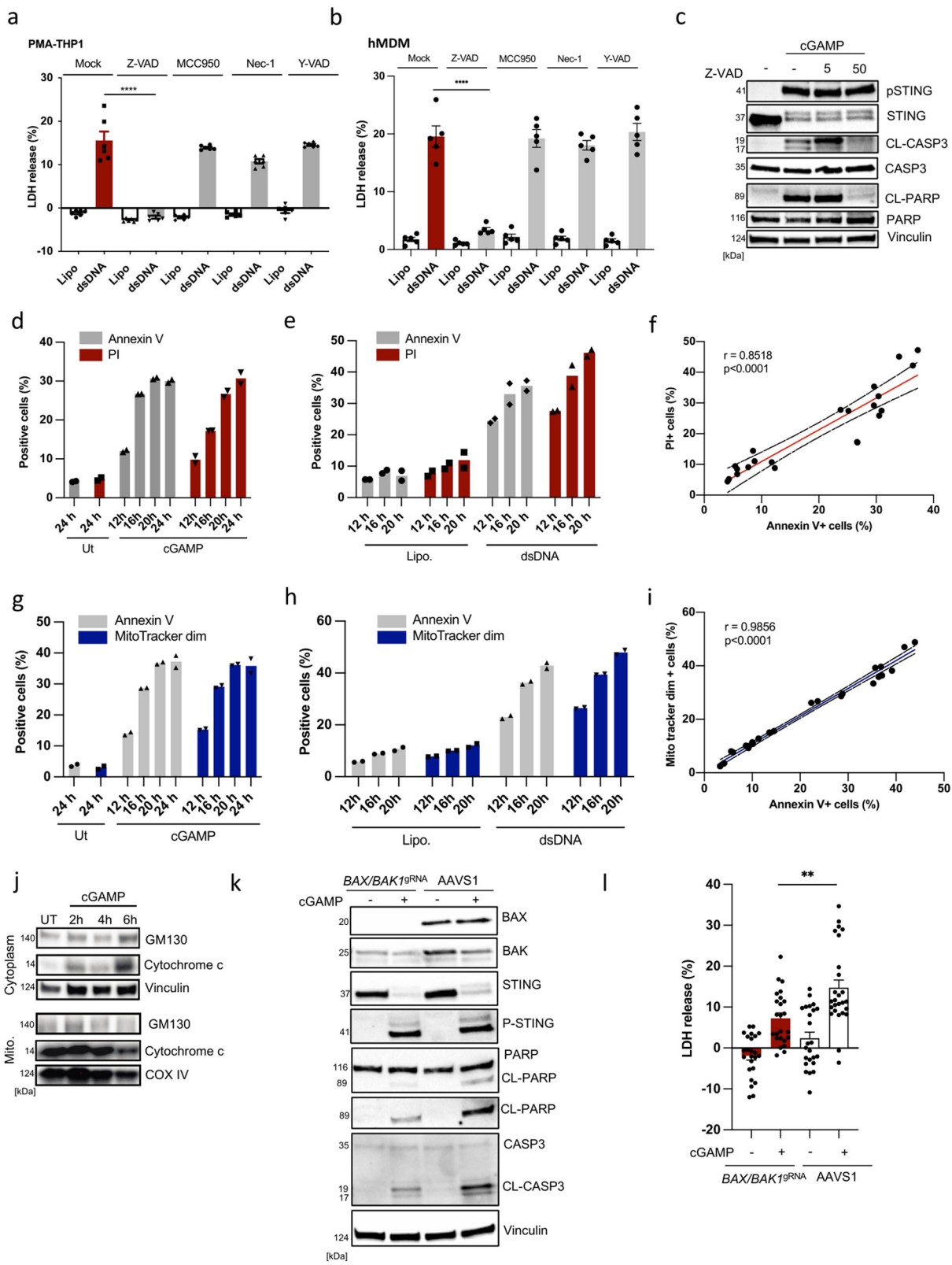

for KHDRBS1, which encodes the protein SAM68, led to impairment of dsDNA- and cGAMP-induced cleavage of caspase 3 and PARP (Fig. 3c, d, Supplementary Fig. S4c, f). The SAM68 dependency was also observed by microscopy of cGAMP-stimulated cells (Fig. 3e, f). Importantly, SAM68-deficient THP1 cells exhibited reduced apoptosis in response to cGAMP and dsDNA (Fig. 3g, h). Reconstituting SAM68 expression in *KHDRBS1*$^{-/-}$

cells through the introduction of KHDRBS1 in vitro-transcribed mRNA by electroporation, or using a lentivirus expressing SAM68, rescued cGAMP-stimulated cleavage of caspase 3 and PARP (Fig. 3i, j). Finally, to examine whether SAM68 plays a role in STING-driven apoptosis in murine macrophages in vivo, we injected WT and *Khdrbs1*$^{-/-}$ mice with cGAMP into the peritoneal cavity. Subsequently, we isolated peritoneal cells for

**Fig. 1 | STING induces cell death through the mitochondrial apoptosis pathway in monocytes/macrophages. a** PMA-differentiated THP1 cells (PMA-THP1) were treated with Z-VAD, MCC950, Nec-1, and Y-VAD at 10 μM for 1 h prior to transfection with dsDNA (4 μg.mL$^{-1}$). LDH release was assessed in the culture supernatants 9 h post dsDNA treatment. Data are the means ± SEM of three independent experiments performed in biological duplicates. **b** Human monocyte-derived macrophages (hMDM) were treated with Z-VAD, MCC950, Nec-1, and Y-VAD at 10 μM for 1 h prior to transfection with dsDNA (4 μg.mL$^{-1}$). LDH release was assessed in the culture supernatants 16 h post dsDNA treatment. The data are the means ± SEM of one experiment performed in five biological replicates from one PBMC donor. **c** PMA-THP1 cells were treated with Z-VAD (5 or 50 μM) for 1 h following exogenous stimulation with cGAMP (100 μg.mL$^{-1}$) for 5 h. Cells were lysed and immunoblotted for the indicated proteins (n = 3). CL cleaved; pSTING (phosphorylated STING). **d–e** PMA-THP1 cells were treated with dsDNA (4 μg.mL$^{-1}$) and cGAMP (100 μg.mL$^{-1}$) for the indicated time intervals and evaluated for staining for Annexin V and Propidium Iodide (PI) by flow cytometry. Data are from one representative experiment performed in biological duplicates. Experiment has been repeated twice with a similar trend. **f** Illustration of data presented in panel D and E as PI$^+$ versus Annexin V$^+$. **g, h** THP1 cells were treated with exogenous cGAMP (100 μg.mL$^{-1}$) or dsDNA (4 μg.mL$^{-1}$) for the indicated time intervals and evaluated for staining of Annexin V and Mitotracker by flow cytometry. Data are from one representative experiment performed in biological duplicates. Experiment has been repeated twice with a similar trend. **i** Illustration of data presented in panel D and E as Mitotracker$^{dim}$ versus Annexin V$^+$. **j** PMA-THP1 cells were treated with exogenous cGAMP (100 μg.mL$^{-1}$) for the indicated time intervals. Cytoplasmic and mitochondrial lysates were monitored for the indicated proteins by Immunoblotting (n = 3). **k, l** THP1 cells were treated with Cas9-gRNA RNP complexes targeting AAVS1 or a combination of BAX and BAK1 gRNA and treated with cGAMP (100 μg.mL$^{-1}$) for 5 h in **k** and 19 h in **l**. Cell survival was monitored by LDH release assay in **l** and immunoblotting for cleavage of Caspase 3 and PARP in **k**. Data are the means ± SEM of four independent experiments performed in biological triplicates or duplicates (**l**). Data are from one representative experiment in **k** that has been performed twice. Statistical analysis of the data in **a**, **b**, and **l** was performed using a two-tailed one-way ANOVA followed by Sidak's multiple comparison test. Vertical stacks of bands are not derived from the same membrane in **c**, **j** and **k**.

evaluation of markers of apoptosis. Interestingly, cGAMP treatment did induce cleavage of caspase 3 in peritoneal macrophages of WT mice, which was partially reduced in cells isolated from cGAMP-treated SAM68-deficient animals (Fig. 3k–m). In further support of cGAMP-driven apoptosis occurring in murine macrophages, we found that treatment of bone-marrow-derived murine macrophages (BMDMs) with exogenous cGAMP led to increased LDH release and cleavage of the apoptotic marker Caspase 3 (Supplementary Fig. S5a, b). The addition of cGAMP exogenously or through lipofection did not alter its capacity to cleave Caspase 3 (Supplementary Fig. S5c). Interestingly, the cleavage of caspase 3 was even further enhanced in BMDMs stimulated with the more stable STING agonizts cGAM(PS)$_2$ and diABZI (Supplementary Fig. S5d). These data demonstrate the broad capacity of STING induction to trigger cell death in murine macrophages of different origins expressing SAM68. Collectively, these data show an essential role for SAM68 in STING-induced apoptosis in both human and murine macrophages.

## SAM68 is not involved in STING-mediated activation of the IFN and NF-κB responses

Type I IFN gene expression is by far the best-studied response to STING signaling[42], but so far, there is limited knowledge on factors governing specificity for the different branches of STING signaling. Interestingly, although SAM68-deficient cells had impaired STING-driven apoptosis, cGAMP-induced expression of *IFNB* expression was not reduced (Fig. 4a). Indeed, the markers of induction and action to the type I IFN pathway (induction: pSTING, pTBK1, pIRF3; action pSTAT1, IFIT1) were not affected or elevated in the absence of SAM68 (Fig. 4b–d; Supplementary Fig. S6a). The parental THP1 cell line in which we disrupted KHDRBS1 has been engineered to integrate two inducible reporter constructs stably. This allows for the simultaneous detection of NF-kB and the IRF promotor activity. SAM68-deficiency led to slightly elevated activation of both promoters in response to cGAMP (Fig. 4e, f). This was paralleled by unaltered or elevated induction of *CXCL10*, *TNFA*, and *IL6* gene expression in SAM68-deficient cells (Supplementary Fig. S6b–d).

## STING interacts with SAM68 at the Golgi to link to the mitochondrial apoptotic machinery

SAM68 has previously been linked to apoptosis through mediating differential splicing of Bcl-x from the long anti-apoptotic to the short pro-apoptotic isoform, and through promotion of TNFα signaling[43,44]. However, in THP1 macrophages, cGAMP did not alter Bcl-x splicing (Supplementary Fig. S7a). Likewise, blockage of TNFα signaling did not affect cGAMP-induced PCD (Supplementary Fig. S7b, c).

To link STING and SAM68 to apoptosis mechanistically, we performed IP of STING and immunoblotted for SAM68. These data showed that SAM68 was indeed recruited to STING following cGAMP treatment,

coinciding with the recruitment of IRF3 and preceding the appearance of the apoptosis marker cleaved PARP (Fig. 5a). At this time point, STING localizes to the Golgi and post-Golgi compartments[19]. We could detect colocalization between SAM68 and STING in cGAMP-stimulated cells. However, the majority of SAM68 has a nuclear location (Fig. 5b). Additionally, the use of Brefeldin A, a drug which blocks COPI-mediated trafficking, abolished both cGAMP-induced activation of antiviral immunity (pSTING, pIRF3) and apoptosis (CL-CASP3 and CL-PARP) (Fig. 5c). Previous work has demonstrated that Golgi-exit of proteins is blocked at 20 °C[45], and we therefore compared induction of apoptosis by cGAMP at 20 versus 37 °C. As shown in Fig. 5d, cGAMP-induced cleavage of caspase 3 and PARP was abolished at 20 °C, whereas phosphorylation of STING and IRF3 was retained, although delayed (Fig. 5d). Subcellular fractionation of cells showed that SAM68 co-fractionated with the Golgi marker GM130 in cGAMP-stimulated THP1 cells (Fig. 5e). Additionally, confocal microscopy supports that STING traffics to the Golgi compartment, where it meets with SAM68 upon cGAMP-stimulation in PMA-THP1 cells (Fig. 5f). This suggests that STING associates with SAM68 at the Golgi and requires exit from this organelle to execute the apoptotic response. Altogether, the data indicates that SAM68 enables STING to execute apoptosis through the intrinsic pathway after exiting from the ER.

## Discussion

Cells respond to the alteration of homeostasis by activating stress, defense, and repair reactions. For instance, excessive protein synthesis leads to ER stress and the unfolded protein response[46], and virus infections lead to the expression of type I IFN with antiviral activity[47]. The ultimate response to acute cellular stress is to induce death, which occurs through different PCD pathways. One potent inducer of apoptosis is cytoplasmic DNA, which occurs during microbial infections, cancer, or autoinflammation[32]. In this work, we have investigated the mechanism of apoptosis in macrophages through the STING pathway, which is activated after cGAMP production by cGAS upon DNA sensing. We report that STING induces apoptosis in macrophages in a manner dependent on IRF3 and TBK1 but independent of transcription and the type I IFN system. Through orthogonal analysis of mass spectrometry datasets, we identified SAM68 as a protein involved in STING-induced cell death but redundant for the IFN response. SAM68 was recruited to STING upon trafficking to the Golgi and served as an executor of mitochondrial apoptosis in macrophages.

SAM68 is expressed in many cell types, with a predominant localization in the nucleus[48]. Here, it promotes alternative mRNA splicing by recognizing RNA sequences adjacent to the included and excluded exons and also facilitates the nuclear export of specific RNA species[48]. For instance, SAM68 modulates alternative splicing of Bcl-x from the Bcl-x(L) to the Bcl-x(S) isoform, thus promoting apoptosis[43]. The RNA-related activities of SAM68 are dependent on the RNA-binding KH domain[43]. In addition to the

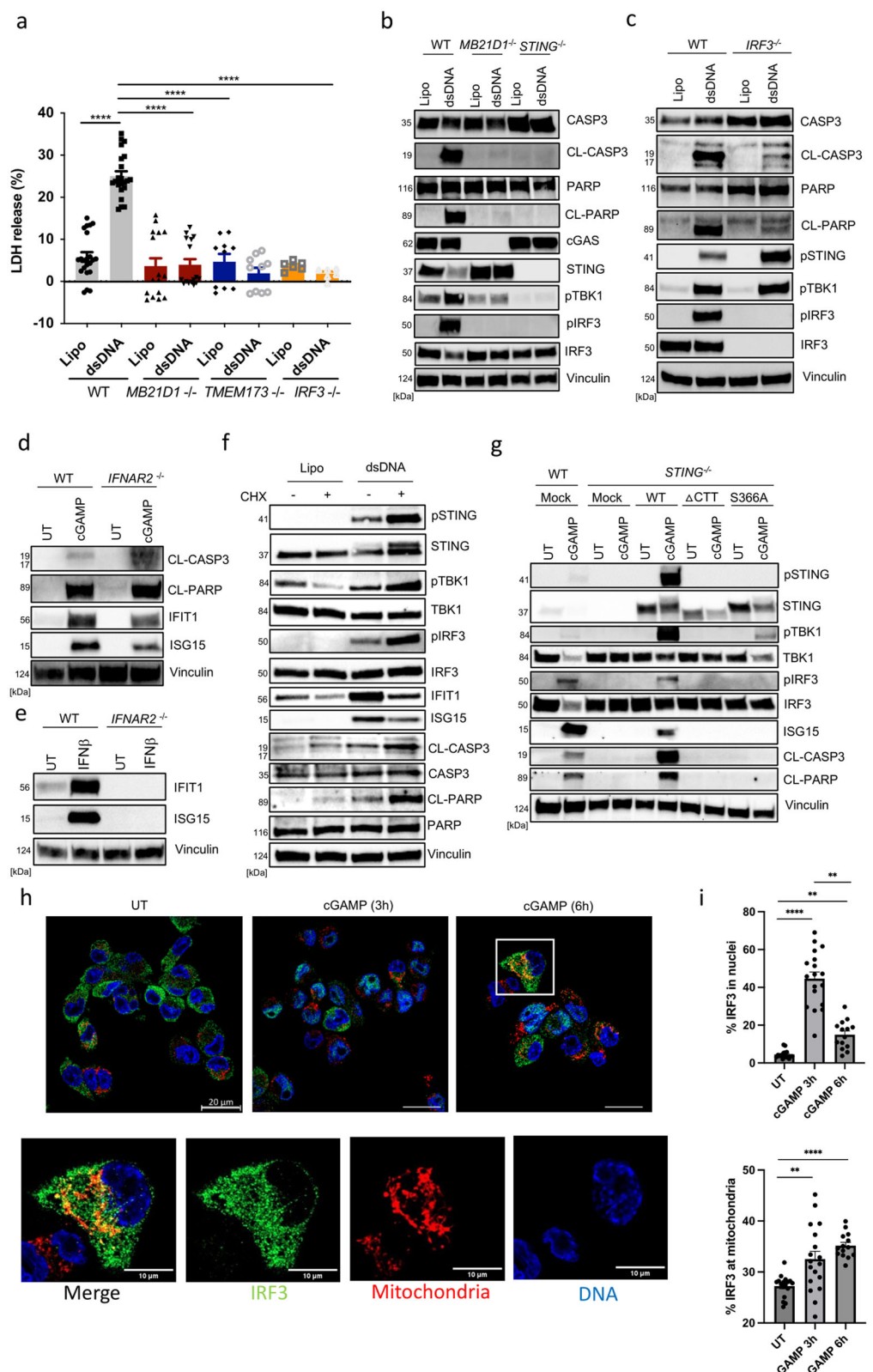

roles in RNA biology, SAM68 has also been ascribed to cytoplasmic activities, mainly in signal transduction. For instance, leptin and insulin receptor stimulation enhances the association of SAM68 with PI3K p85 subunit to promote PI3K activity[49,50]. Also, SAM68 has been demonstrated to play a role in signaling from the TNF receptor to NF-κB[44,51]. In addition, the role of SAM68 in promoting TNF signaling has also been reported to augment TNF-induced apoptosis[44,51]. In our hands, STING activation did not stimulate alternative splicing of BCL-x, nor was the cGAMP-induced apoptotic response dependent on TNFα. Instead, SAM68 interacted with STING following its trafficking and facilitated the execution of intrinsic apoptosis. Hence, this work provides yet another mechanism through which SAM68 directs STING signaling toward apoptosis in macrophages.

**Fig. 2 | STING operates through IRF3 to trigger apoptosis in THP1 macrophages. a** Parental, cGAS KO, STING KO, and IRF3 KO PMA-THP1 cells were transfected with dsDNA (4 μg.mL⁻¹). LDH release was assessed in the culture supernatants 9 h post dsDNA treatment. Data are the means ± SEM of four independent experiments performed in four to six biological replicates. **b–d** WT, $cGAS^{-/-}$, $STING^{-/-}$, $IRF3^{-/-}$, $TBK1^{-/-}$, and $IFNAR2^{-/-}$ PMA-THP1 cells were treated with dsDNA (4 μg.mL⁻¹) or cGAMP (100 μg.mL⁻¹) for 5 h, and lysates were immunoblotted for the indicated protein (*n* = more than 3 for most of the KO cell lines tested). CL cleaved, pSTING phosphorylation at S366, pTBK1 phosphorylation as S172, pIRF3 phosphorylation at S396. **e** WT and $IFNAR2^{-/-}$ PMA-THP1 cells were treated with IFNβ (100 units/mL) for 1 h, and lysates were immunoblotted for the indicated protein (*n* = 2). **f** PMA-THP1 cells were treated with cycloheximide (CHX) (1 μg.mL⁻¹) for 1 h and transfected with dsDNA (4 μg.mL⁻¹). Lysates isolated 3 h post dsDNA transfection were immunoblotted for the indicated protein (*n* = 3). pIRF3, phosphorylated IRF3 (S396). CL, cleaved. **g** STING-deficient THP1 cells were transduced with lentivirus encoding WT, ΔCTT or S366A STING. The cells were treated with cGAMP (100 μg.mL⁻¹) for 3 h and lysates were immunoblotted for the indicated proteins (*n* = 2). CL cleaved, p phospho. **h** PMA-THP1 cells were treated with cGAMP (100 μg.mL⁻¹) for the indicated period of times, and stained for DNA, IRF3 and mitochondria. Cells were visualized by confocal microscopy (*n* = 3). Scale bars, 20 μm. White box indicates the region of interest which was selected by the rectangular function in ImageJ and displayed as separate images, the scalebar corresponds to 10 μm. **i** Quantification of the percentage of IRF3 area within nuclei or mitochondria from **h**. Data are the means ± SEM of three independent experiments. Statistical analysis of the data in **a** and **i** was performed using a two-tailed one-way ANOVA followed by Sidak's multiple comparison test. Vertical stacks of bands are not derived from the same membrane in **b–g**.

DNA can induce several modalities of PCD, including apoptosis, lysosomal cell death, autophagic cell death, necroptosis, and pyroptosis[32]. For the cGAS-STING pathway, there seems to be a significant cell-type specificity regarding the modality and mechanism of PCD. For instance, mouse T cells undergo apoptosis through the STING pathway via IRF3-driven transcription of the apoptosis genes Noxa and PUMA[28]. In the same study, the authors also demonstrated that the murine STING agonist CMA did not induce PCD in bone-marrow-derived macrophages (BMDMs). However, our findings indicate that cGAMP stimulation of BMDMs does lead to PCD as assessed by LDH release and cleavage of caspase 3. This discrepancy could be because of the different types of STING agonizts used or potential differences in the BMDM protocols. Other studies in fibroblasts and BMDMs have also demonstrated transcriptional programs involved in STING-driven cell death in murine cells, which was effectuated by TNF-dependent necroptosis[30,52]. Human monocytes also undergo STING-driven cell death via IRF3-mediated signaling to the mitochondria independent of transcription[31]. In agreement, we previously reported cGAS-dependent apoptosis of microglia/macrophage in the herpes simplex virus-infected mouse brain to be dependent on IRF3 but independent of IFNAR[29]. Aside from STING's capacity to trigger apoptosis through IRF3 transcriptional dependent and independent mechanisms, Gaidt et al. reported that STING activation can engage a lytic lysosomal cell death program (LCD) in myeloid cells, operating independently of all other known canonical PCD pathways[25]. Mechanistically, this STING-induced death program was found to operate independently of TBK1, IκB kinase-ε (IKKε), and IRF3. Although STING-activated BLaER1 cells expressed some signatures of apoptosis, including activation and cleavage of caspase-3, the pre-treatment of these cells with a pan-caspase inhibitor was not able to impair the LCD program. Contradictory, using PMA-THP1 and primary human MDMs, we demonstrated that STING-induced cell death is dependent on IRF3 and only the pan-caspase inhibitor Z-VAD and not the Caspase 1, NLRP3, or necroptosis inhibitors could prevent STING-induced cell death as measured by LDH release. However, it should be noted that the cellular models used in these two studies are significantly different, and the PCD literature contains numerous examples of cell-type specificity regarding the modality and mechanism of death. Hence, the data presented in this work provide insight into the mechanism of STING-dependent cell death in myeloid cell lines and primary monocyte-derived macrophages. It is also important to stress that our work has focused on the immediate signaling from STING to apoptosis. It is likely that the STING-induced gene expression program, which includes genes with pro-apoptotic activity[53], may also contribute to the apoptotic response at later time-points.

The cGAS-STING pathway activates several downstream effector functions. This includes expression of type I/III IFN genes, induction of NF-κB-induced genes, autophagy, and apoptosis[32]. All of these activities are dependent on STING trafficking from the ER to the ERGIC/Golgi[54], which is a rate-limiting step in STING signaling[11,55]. However, there is limited knowledge of the determinants governing specificity in STING signaling towards the different effector activities. At the cellular level, it is known that activation of IFN occurs at the ERGIC/Golgi compartment where STING phosphorylation at Serine 366 occurs and IRF3 is recruited[12,56]. The ERGIC/Golgi compartment also serves as a membrane source for LC3 lipidation in STING-driven autophagy[15]. The subcellular compartment for NF-κB activation remains unknown. Our data suggest that STING activation drives IRF3 translocation to the mitochondria, triggering BAX/BAK1-dependent apoptosis. Although the effect of the double KD BAX/BAK1 is incomplete, the cellular phenotype is consistent with the rest of the data presented, and support the conclusions drawn. One explanation for this observed partial phenotype could be that we did not work with KO cell clones but rather with a bulk population of cells treated with gRNAs targeting two genes. Not all cells from the bulk are fully KO for both proteins and this could explain some of the residual activity seen in terms of Caspase3 and PARP cleavage. Alternatively, another pathway than the BAX/BAK1 mitochondrial pathway could also be at play in cGAMP-treated cells leading to the observed residual cleavage of Caspase3 and PARP.

Our work also demonstrates that STING-induced apoptosis occurs post-Golgi, since we observed that cGAMP-induced apoptosis was impaired at 20 °C, whereas cGAMP-induced IFN responses were retained. In addition, we found STING to co-fractionate and colocalize with SAM68 and the Golgi marker GM130. These data suggest that STING associates with SAM68 at the Golgi and could help as a relay to drive the intrinsic apoptosis pathway upon exit from the Golgi. Interestingly, SAM68 deficiency did not repress IFN responses and activation of NF-κB, thus showing that SAM68 acts specifically in the apoptosis pathway. STING has also been demonstrated to be sorted into autophagosomes and lysosomal degradation pathways as part of the termination of signaling[19,57]. Since this occurs post-Golgi, exploring the interaction between the STING-SAM68-apoptosis and STING-degradation pathways will be interesting.

Regarding the pathophysiological relevance of the present finding, several pieces of data have demonstrated the importance of STING-driven apoptosis in myeloid cells. For instance, the inflammatory disease STING-associated vasculopathy with onset in infancy (SAVI), a monogenic disease driven by gain-of-function mutations in STING, is associated with extensive leukopenia and spontaneous cell death in monocytes from the patients[33]. In addition, we reported that microglia/macrophages undergo cGAS-dependent cell death in the brain to down-regulate inflammatory response during viral infection[29]. More work is needed to establish whether SAM68 plays a role in these phenomena. Interestingly, Sam68 deficient mice develop significantly less severe inflammation than wildtype mice in an experimental colitis model, which is dependent on STING signaling and where macrophage apoptosis ameliorates symptoms[58–61]. These data provide an in vivo link between SAM68-dependent apoptosis, STING, macrophages, and disease outcome and warrant further investigations. In summary, we report SAM68 as a mechanistic link between STING activation and induction of apoptosis through the intrinsic pathway in macrophages.

## Experimental procedures
### Cell lines and culture conditions
WT THP1 cells (ATCC) and THP1-Dual™ cells (hereafter called THP1 cells, Invivogen) derived from the human THP1 monocytic cell line

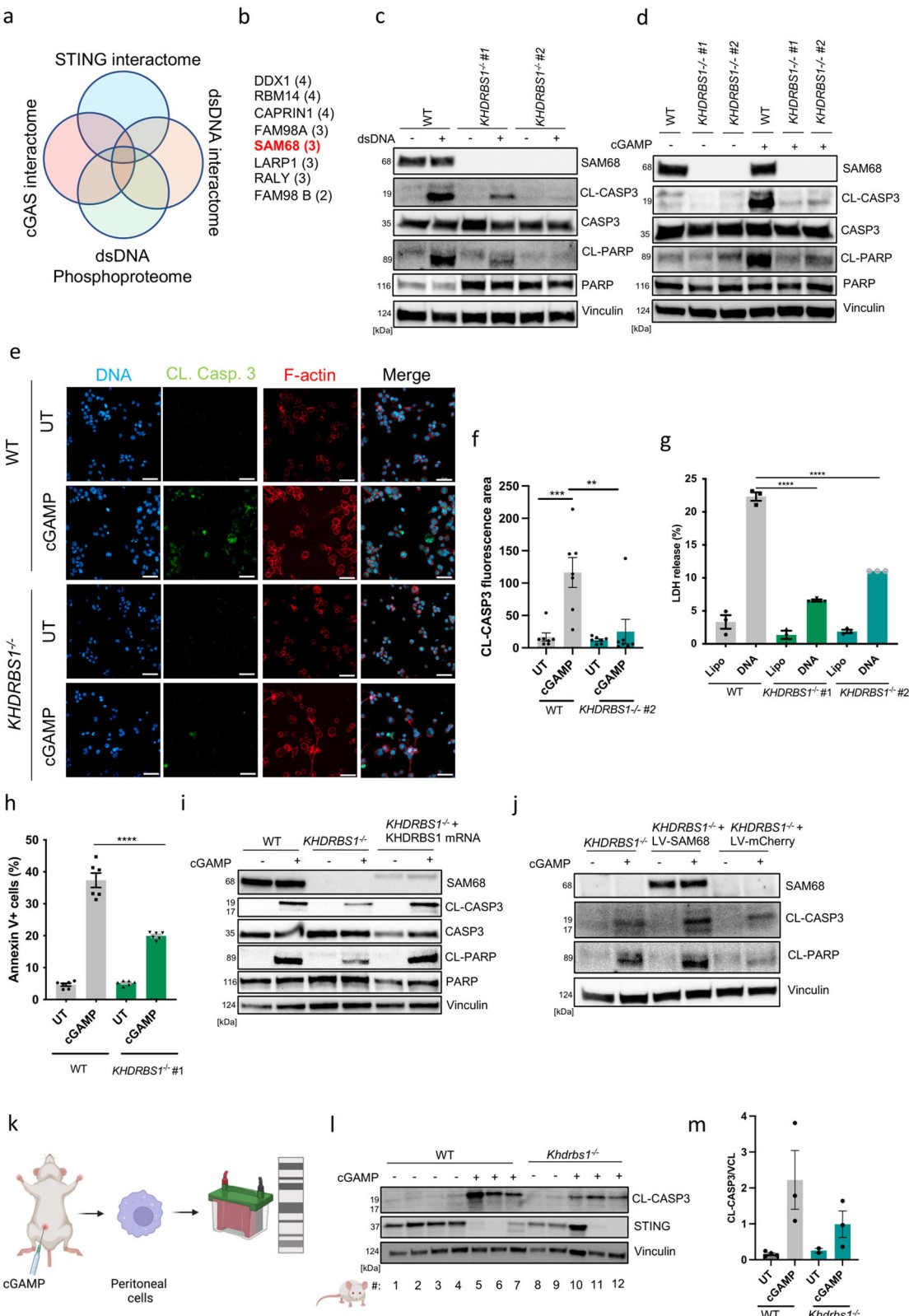

were grown at 37 °C (5% $CO_2$) in RPMI-1640 medium (Sigma-Aldrich) supplemented with heat-inactivated 10% fetal calf serum (FCS), 200 IU/mL penicillin, 100 µg/mL streptomycin, and 600 µg/mL glutamine (hereafter called RPMI complete). THP1 cells were differentiated into macrophages (hereafter called PMA-THP1) by stimulation with 150 nM phorbol 12-myristate 13-acetate (PMA, Sigma-Aldrich) in RPMI complete medium for

24 h. After PMA-stimulation the medium was refreshed, allowing the cells to further differentiate for another 24 h.

### Knock-out (KO) cell lines

THP1-Dual™ KO-cGAS cells, THP1-Dual™ KO-STING cells, THP1-Dual™ KO-IRF3 cells, THP1-Dual™ KO-TBK1 cells, THP1-Dual™ KO-

**Fig. 3 | SAM68/KHDRBS1 is essential for induction of apoptosis downstream of STING. a** Venn diagrams illustrating data sets included in the screen and selection criteria. **b** Proteins fulfilling the selection criteria. number in brackets, indicate the number of data sets in which the individual proteins were identified. **c, d** WT and *KHDRBS1*$^{-/-}$ (SAM68) PMA-THP1 cells (two different clones) were treated with dsDNA (4 μg.mL$^{-1}$) or cGAMP (100μg.mL$^{-1}$) for 5 h, and lysates were immunoblotted for the indicated proteins. (*n* = 3). CL cleaved. **e** WT and *KHDRBS1*$^{-/-}$ PMA-THP1 cells were treated with cGAMP (100 μg.mL$^{-1}$) for 5 h, fixed, and were stained for cleaved caspase 3 (CL-CASP3) and visualized by confocal microscopy (*n* = 2). Scale bars, 50 μm. **f** Quantification of data from panel **e**. Pixel density from the CL-CASP3 staining was evaluated using ImageJ software. The data are represented as total CL-CASP3 immunofluorescent area. Data are the means ± SEM of two independent experiments. **g** WT and *KHDRBS1*$^{-/-}$ PMA-THP1 cells were stimulated with dsDNA (4 μg.mL$^{-1}$) for 9 h. Levels of cell death were evaluated by LDH release assay. Data are the means ± SEM of one representative experiment performed in biological triplicates. Experiment has been repeated twice with a similar trend. **h** WT and *KHDRBS1*$^{-/-}$ THP1 cells were stimulated with cGAMP (100 μg.mL$^{-1}$) for 18 h. Levels of apoptosis were evaluated by annexin V staining and flow cytometry. Data

are the means ± SEM of two experiments performed in biological triplicates. **i** *KHDRBS1*$^{-/-}$ (SAM68) THP1 cells were transfected with mRNA encoding SAM68 and subjected to PMA-differentiation in parallel with WT and *KHDRBS1*$^{-/-}$ THP1 cells. The cells were stimulated with cGAMP (100 μg.mL$^{-1}$) for 5 h, and evaluated for the specific proteins by immune blotting. (*n* = 2). CL cleaved. **j** *KHDRBS1*$^{-/-}$ (SAM68) THP1 cells were transduced with a lentivirus encoding for SAM68 and subjected to PMA-differentiation. Cells were stimulated with cGAMP (100 μg.mL$^{-1}$) for 5 h, and evaluated for the specific proteins by immunoblotting. (*n* = 2). CL cleaved. **k** Schematic representation of the work layout. Animals were injected intraperitoneally with 2'3'-cGAM(PS)2 (Rp/Sp) (125 μg/mouse). Following cGAMP administration (8 h), peritoneal cells were collected and subjected to immunoblotting. Each line represents an individual animal. This figure was created using BioRender.com. **l** Peritoneal cells from cGAMP-injected animals were resolved on SDS-PAGE and subjected to immunoblotting. Each line represents an individual animal. **m** Quantification of CL-CASP3 protein levels presented in **l**. Statistical analysis of the data in **f–h** was performed using a two-tailed one-way ANOVA followed by Sidak's multiple comparison test. Vertical stacks of bands are not derived from the same membrane in **c, d, i, j** and **l**.

IFNAR2 cells, THP1-Dual™ KO-ASC cells (Invivogen) were used to study STING-induced cell death. Also, SAM68 KO cells were generated from THP1-Dual™ cells using CRISPR-Cas9 technology (see dedicated section). All cell lines were cultured as described above.

### Genome editing in THP1 cells using CRISPR-Cas9 strategy and gene KO validation

In this work, KO cell lines were generated either as pools of KO cells or as single SAM68 KO clones. In brief for pool generation of single KO cells, WT THP1 cells (1-2 × 10$^5$ cells) in 15 μl of Opti-MEM were incubated at room temperature for 25 min with a 5-μl Opti-MEM solution containing 6 μg of Cas9 Nuclease V3 protein (IDT) and 3.2 μg of modified synthetic gRNA (Synthego). For the generation of BAK1 and BAX double KO cells, cells were electroporated with 6 μg of Cas9 Nuclease V3 protein (IDT) and 2 × 3.2 μg of modified synthetic gRNA (Synthego). After incubation, 20 μl of solution containing cells, Cas9 protein, and gRNAs was transferred to one well of a Nucleocuvette strip and subsequently subjected to electroporation using a 4D Nucleofection machine (Lonza). P3 Primary Cell Nucleofector Solution and pulsing code CM 138 were used to achieve maximal efficiency. After electroporation, 100 μl of prewarmed medium was added to each well. Cells were then transferred to regular tissue culture plates and expanded for at least 3 to 4 days before further use. For clonal KO cell generation, the same general protocol was applied to WT THP1-Dual™ cells. During the RNP complex formation, GFP mRNA (1 μg) was added to the mixture. Following electroporation, cells were single sorted based on GFP expression levels and subsequently grown in complete medium. A total of 96 wells containing single cells were sorted. After 2-3 months of culturing, KO clones were tested for the absence of SAM68 proteins. Two of the clones selected are used throughout the study and were generated using the SAM68 gRNA#2. The sequences of all guide RNAs used are shown in Supplementary Table 1.

The efficacy and identity of the induced genetic modification was assessed 3 days post electroporation using Interference of CRISPR edits (ICE) analysis. In brief, genomic DNA (gDNA) was isolated from ~100.000 cells using QuickExtract™ DNA Extraction Solution (LGC Biosearch Technologies), and PCR amplicons were generated using the 2X Phusion Plus PCR Master Mix (Thermo Fisher Scientific) according to the general description by the manufacturer. The primer list is shown in Supplementary Table 2. Correct amplicon size and purity was confirmed by agarose gel electrophoresis. Primers and incomplete DNA were removed by treatment with Exonuclease I (Thermo Fisher Scientific) and dephosphorylation was performed using FastAP Thermosensitive Alkaline Phosphatase (Thermo Scientific). Sequencing was performed by Eurofins and ICE analysis performed using the ICE software tool by Synthego: https://ice.synthego.com.

### Primary monocyte-derived macrophages

Buffy coats from Aarhus University Hospital blood bank were used to collect peripheral blood mononuclear cells, and monocytes were separated from PBMCs by plastic adherence in RPMI 1640 supplemented with 10% AB-positive human serum (Invitrogen), macrophage colony-stimulating factor (Sigma, 15 ng/mL), glutamine (600 μg/mL), penicillin (200 IU/mL), and streptomycin (100 μg/mL) (all from Gibco). Monocytes were allowed to differentiate into monocyte-derived macrophages for 7 days.

### Bone marrow-derived macrophages (BMDMs) preparation

Animals dedicated for bone marrow extraction were housed at Aarhus University, Department of Biomedicine, under license 2023-15-0201-01489. Bone marrow was extracted from the femur and tibia of 10–14-week-old female C57BL/6BomTac mice (Taconic, Denmark) as described in ref. 62. For macrophage differentiation, bone marrow cells were incubated with 30% of L929 supernatant in DMEM containing 10% FBS, Penicillin and Streptomycin, and 2 mM L-Glutamine in Petri dishes for 7 days. For cGAMP stimulation, BMDMs were replated and stimulated as described below.

### Cell death inhibitors

If not otherwise specified, one of the following cell death inhibitors was added to the cells: (10 μM) 1–2 h prior to cell stimulation with dsDNA: Z-VAD(OMe)-FMK (Cell Signaling), Z-DEVD-FMK (R&D Systems), MCC950 (Invivogen), Nec-1 (Abcam) or Ac-YVAD-cmk (Invivogen).

### Other reagents

Anti-TNFα antibody (Cell Signaling, Human TNF-α Neutralizing (D1B4) Rabbit mAb); 2'3' cGAMP and 2'3'-cGAM(PS)2 (Rp/Sp) and diABZI (all Invivogen), and TNFα (Abcam, ab259410); Mito tracker (ThermoFisher); ABT-737 and S63845 (MedChemExpress); A listing of all Antibodies used in the study is appended to Supplementary Table 3.

### In vivo treatment with STING agonizts

C57BL/6J mice wild-type and Sam68−/− female mice were maintained on a normal 12 h light/dark cycle in the animal facility of the Fondazione Santa Lucia IRCCS and genotyped by the Biotool Mouse Direct PCR Kit. Mice breeding, housing and treatments were conducted according to the Guidelines of the Italian Institute of Health complying with all relevant ethical regulations for animal use and approved by the protocol n. 157/2019-PR. WT and KO mice were injected intra-peritoneally with 2'3'-cGAM(PS)$_2$ (Rp/Sp) (125 μg/mouse). Peritoneal lavage was performed 8 later, and the isolated cells were lysed in 10X RIPA Buffer supplemented with protease inhibitors, for subsequent analysis by immunoblotting.

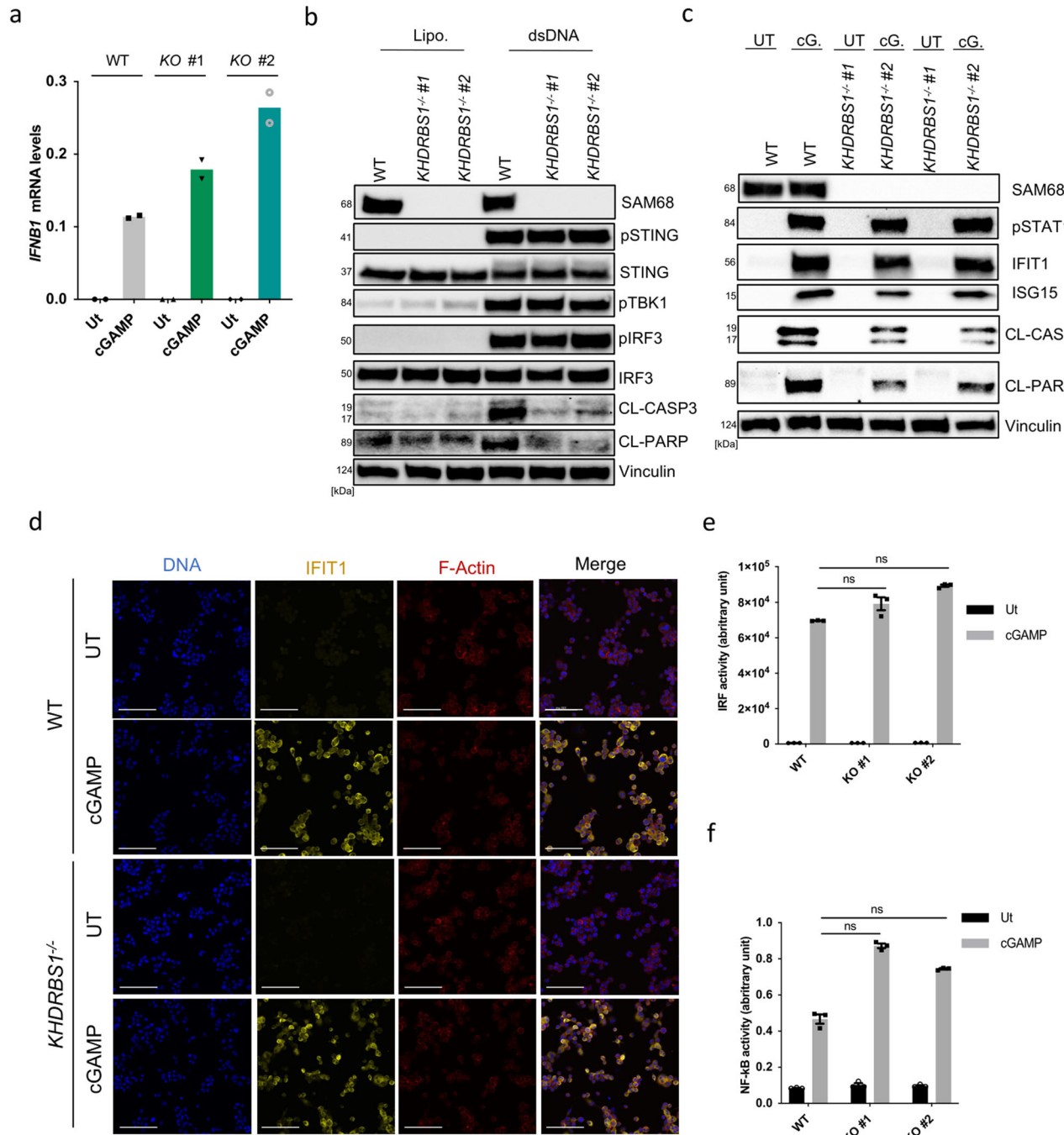

**Fig. 4 | SAM68 is not involved in STING-dependent activation of the IFN and NF-κB responses. a** WT and *KHDRBS1*$^{-/-}$ PMA-THP1 cells were treated with cGAMP (100 μg.mL$^{-1}$) for 5 h. RNA was isolated and examined for levels of *IFNB1*. The data are the means of one experiment performed in biological duplicates. **b, c** WT and *KHDRBS1*$^{-/-}$ PMA-THP1 cells were treated with dsDNA (4 μg.mL$^{-1}$) and cGAMP (100 μg.mL$^{-1}$) for 5 h and 18 h, respectively. Lysates were immunoblotted for the indicated proteins (*n* = 3). pSTING phosphorylation at S366, pTBK1 phosphorylation as S172, pIRF3 phosphorylation at S396, pSTAT1 phosphorylation at Y701. **d** WT and *KHDRBS1*$^{-/-}$ THP1 cells were treated with cGAMP

(100 μg.mL$^{-1}$) for 5 h, fixed stained with antibodies against IFIT1 and actin, and visualized by fluorescence microscopy (*n* = 1). Scale bars, 100 μm. **e, f** WT and *KHDRBS1*$^{-/-}$ PMA-THP1 cells were treated with cGAMP (100 μg.mL$^{-1}$) for 5 h, IRF and NF-κB gene activities were measured as reported in the methodological section (*n* = 3). Statistical analysis of the data performed on means ± SEM in **e** and **f** and assessed using a two-tailed one-way ANOVA followed by Sidak's multiple comparison test. Vertical stacks of bands are not derived from the same membrane in **b** and **c**.

## DNA transfection and cGAMP treatment of cells

An HSV-60 Naked viral dsDNA motif (Invivogen) was delivered intracellularly at a final concentration of 4 μg/ml using Lipofectamine 2000 Transfection Reagent (Invitrogen) diluted in unsupplemented serum-free RPMI-1640 medium at a Lipofectamine/dsDNA ratio of 1:1. STING

agonist 2'3'-cGAMP (Invivogen) was delivered extracellularly at a final concentration of 50–100 μg.mL$^{-1}$ or was lipofected using lipofectamine 2000 (10 μg.mL$^{-1}$). STING agonists 2'3'-cGAM(PS)$_2$ and diABZI (Invivogen) were delivered extracellularly at a final concentration of 12.5–100 μg.mL$^{-1}$.

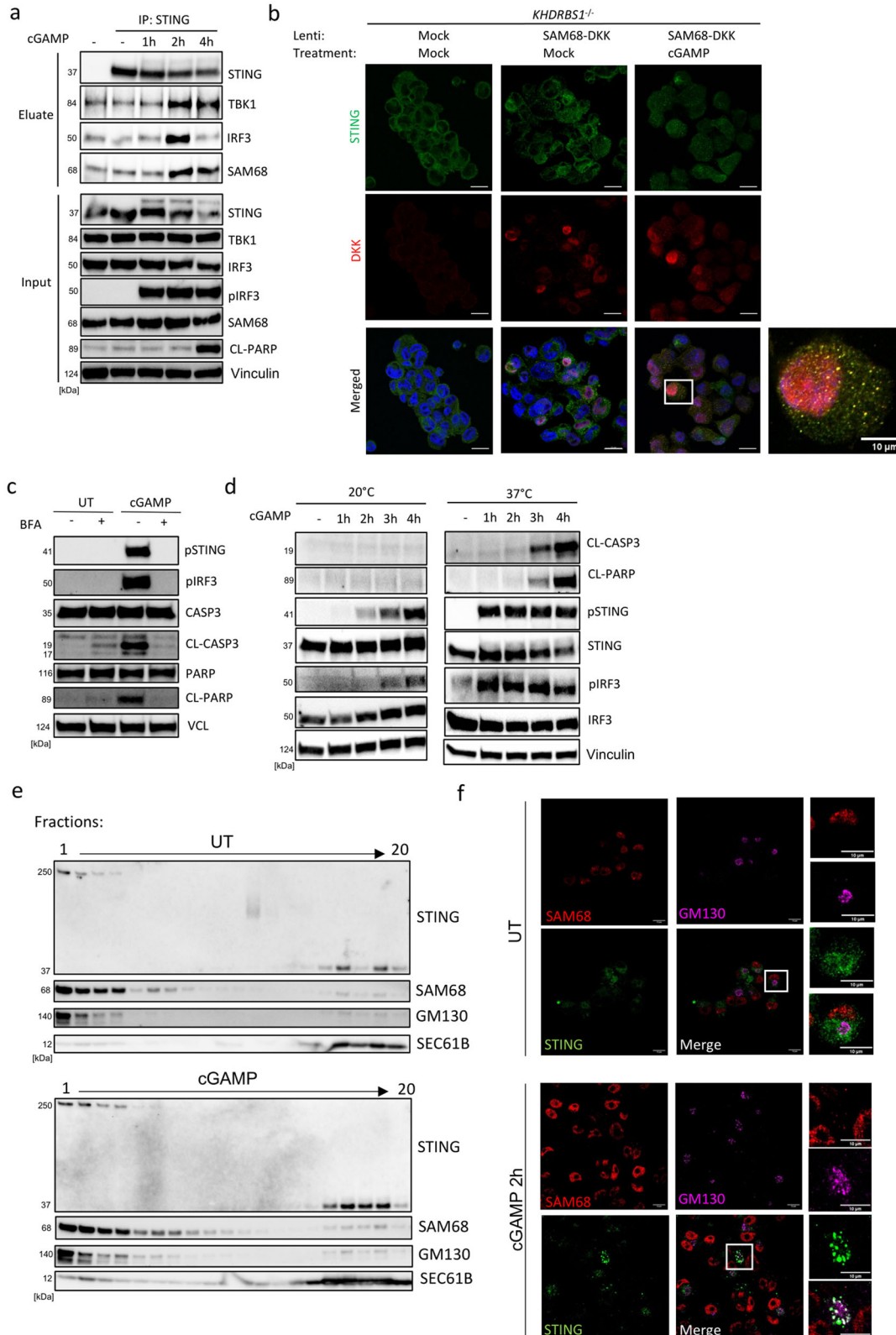

## Flow cytometry

The Dead Cell Apoptosis Kit with Annexin V FITC and PI (Invitrogen) was used to detect apoptosis by flow cytometry. Apoptosis was induced by dsDNA transfection or cGAMP stimulation, as described above. Untreated cells served as a negative control. Harvested cells were washed in cold PBS and resuspended in 1× Annexin-binding buffer. To stain the cells, 5 µl of AlexaFluor488 Annexin-V and 1 µl of 100 µg/ml PI solution were added per 100 µl of cell suspension and incubated for 15 min at RT. Cells were analyzed by flow cytometry measuring fluorescence emission at 530 nm and >575 nm using a NovoCyte Flow Cytometer (ACEA Biosciences). Data were analyzed using FlowJo software v10.6.2 (FlowJo LLC). The gating strategy is displayed in Supplementary Fig. S8.

**Fig. 5 | SAM68 links the STING-IRF3 complex to the mitochondrial apoptosis pathway. a** PMA-THP1 cells were treated with cGAMP (100 µg.mL⁻¹) for 1, 2, and 4 h, and lysates were subjected to immunoprecipitation with anti-STING antibodies and immunoblotted for the indicated proteins (*n* = 2). CL, cleaved. **b** KHDRBS1-deficient PMA-THP1 cells reconstituted with a lenti-DKK-tagged SAM68 were treated with cGAMP (100 µg.mL⁻¹) for 5 h, fixed, stained with anti-STING and anti-DKK, and visualized by fluorescence microscopy (*n* = 2). Scale bars, 20 µm. White box indicates the region of interest which was selected by the rectangular function in ImageJ and displayed as separate images, the scalebar corresponds to 10 µm. **c** PMA-THP1 cells were pre-treated with Brefeldin A (BFA) 3 µg.mL⁻¹ for 30 min prior to cGAMP challenge (100 µg.mL⁻¹) for 5 h. Cell lysates were immunoblotted for the indicated proteins (*n* = 2). CL, cleaved. pSTING, phosphorylation at S366; pIRF3,

phosphorylation at S396. **d** PMA-THP1 cells were treated with cGAMP (100 µg.mL⁻¹) at 20 °C or 37 °C for 1, 2, 3, and 4 h, and lysates were immunoblotted for the indicated proteins (*n* = 2). CL, cleaved. pSTING, phosphorylation at S366; pIRF3, phosphorylation at S396. **e** ER- and Golgi-enriched pellets from mock- or cGAMP-treated cells were fractionated and immunoblotted for STING, SAM68, GM130 (Golgi) and Sec61B (ER) (*n* = 2). **f** WT PMA-THP1 cells were treated with cGAMP (100 µg.mL⁻¹) for 2 h, and stained for STING, SAM68 and the Golgi marker GM130. Cells were visualized by confocal microscopy (*n* = 3). Scale bars, 10 µm. White box indicates the region of interest which was selected by the rectangular function in ImageJ and displayed as separate images, the scalebar corresponds to 10 µm. Vertical stacks of bands are not derived from the same membrane in **a**, **c**, **d** and **e**.

## Immunoblotting

1,2 × 10⁶ THP1 cells were washed in PBS and subsequently lysed in 120 µl ice-cold Pierce RIPA lysis buffer (Thermofisher Scientific), supplemented with 1x Complete™ protease inhibitor cocktail (Roche) and 5 IU/mL benzonase (Sigma). Protein concentration was measured using a Pierce™ bicinchoninic acid (BCA) Protein Assay Kit (Thermofisher Scientific), according to manufacturer's instructions. Equalized samples were resuspended in loading buffer, consisting of XT Sample Buffer (BioRad) and XT Reducing Agent (BioRad), and denatured at 95 °C for 3 min. 10–40 µg of total protein from the cell lysate was separated by SDS-PAGE on a 4–20% Criterion™ TGX™ Precasted Gel (BioRad). The gel was first run at 70 V for 20 min, following 110 V for 50 min. Transfer onto a Midi Format 0,2 µM PVDF membrane (BioRad) was done using a Transblot Turbo Transfer System (BioRad) for 7 min. Membranes were blocked for 1 h at room temperature (RT) using 5% skimmed milk (Sigma-Aldrich) diluted in PBS supplemented with 0.05% Tween-20 (PBS-T). Membranes were divided into smaller fragments and incubated overnight at 4 °C with one of the following primary antibodies (see Supplementary Table 3). All antibodies were diluted 1:1000 in PBST and 0.02% sodium azide was added as a preservative. Membranes were washed 3 times in PBST for 15 min following incubation with secondary antibodies: HRP conjugated F(ab)2 donkey anti-rabbit IgG (H + L) or HRP conjugated F(ab)2 donkey anti-mouse IgG (H + L) (1:10.000) (Jackson Immuno Research) in PBST 1% milk for 1 h at RT. Membranes were washed 3 times in PBST for 10 min and subsequently incubated with SuperSignal™ West Dura Substrate or SuperSignal™ West Femto Maximum Sensitivity Substrate (ThermoFisher Scientific) for 1 min prior to exposure using a ChemiDoc Imaging System (BioRad) or a Invitrogen iBright FL1500. The listing of the antibodies used is shown in Supplementary Table 3.

Vertical stacks of bands are not derived from the same membrane; each membrane was divided into smaller fragments and each piece of membranes incubated overnight at 4 °C with one of the following primary antibodies. In the presented Western blot images, molecular weight markers were always run alongside the samples. Uncropped images of the western blots are provided in Supplementary Fig. 9. The molecular weights of observed bands were determined by referencing the positions of bands from the molecular weight markers (MWM) run in parallel. However, MWM are not displayed in some of the supplementary unprocessed figures. Indeed, most of the immunoblotting (IB) data were acquired in 2019–2020 and storage of unprocessed data lived up to NPG standards at the time. All immunoblotting data produced more recently do contain all the information requested by the journal displaying both the protein of interest and the MWM. We do apologize for the lack of some of the size markers, but we are confident that the approach used does not compromise the accuracy and reliability of protein size determination in our experiments.

## Cytochrome c release assay

20 × 10⁶ THP1 cells were seeded in T75 cell culture flask and differentiated with 150 nm PMA in PRMI-1640 medium with 10% FCS, penicillin, and streptomycin. The following day, the cell culture media was refreshed with

10 mL of complete RPMI-1640 medium without PMA. On day 3, the cells were stimulated with 100 µg.mL⁻¹ exogenous cGAMP in kinetics (0, 2, 4, 6 h). The cytosolic and mitochondria fractions were extracted according to the manufacturer's guidelines (Abcam) (Cytochrome c releasing apoptosis assay kit, ab65311).

## Sample preparation for mass spectrometry (MS) analysis of STING-interacting proteins

Protein samples, 4 biological replicates, in PBS with 1% SDS and inhibitors were sonicated on ice using a probe sonicator for 5 cycles (one cycle was comprised of 10 s on and 10 s off) before centrifugation at 17,000 g for 30 min. Supernatant protein concentration measurements were made using a standard BCA assay. Filter-aided sample preparation was performed as previously described[63]. Briefly, protein samples were loaded onto spin units (Vivacon 500, 30 K Da cutoff) and washed three times with UA buffer (8 M urea in 100 mM tetraethylammonium bromide (TEAB)) to remove SDS. Subsequently, proteins were reduced by adding 50 µL of 50 mM dithiothreitol (DTT) in UA buffer and incubating at 37 °C for 1 h. After centrifugation to remove excess DTT, proteins were then alkylated by adding 50 µL of 50 mM iodoacetamide (IAA) in UA buffer and incubating at room temperature for 20 min in the dark. After centrifugation to remove excess IAA and one more wash by UA buffer, 20 µL of Lys-C in UA buffer was added (enzyme to protein ratio at 1:100). The spin unit was then incubated at 37 °C for 3 h, followed by addition of 200 µL of trypsin in 100 mM TEAB (enzyme to protein ratio at 1:50) and incubation at 37 °C overnight. Resultant peptides were collected from the spin unit by centrifugation and 0.1% of trifluoroacetic acid added to deactivate any remaining enzymes. Samples were then vacuum dried and stored at −80 °C until MS analysis.

## Liquid chromatography coupled with tandem mass spectrometry (LC-MS/MS) analysis of STING-interacting proteins

Peptide samples were reconstituted in 0.1% of formic acid and subjected to duplicate LC-MS/MS analysis by a QExactive mass spectrometer (Thermo Fisher) coupled to an easy nLC-1000 (Thermo Fisher). A 60-minute gradient with a 50-minute separation window (5–22% acetonitrile in 0.1% formic acid in 40 min then to 40% in 10 min) was used to separate peptides. Data dependent acquisition with a loop count of 10 was performed. Precursor scan were performed as follows: scan range 300–1800, resolution of 70,000, automatic gain control (AGC) of 1e6 and maximum injection time of 20 ms. Fragment scan was done as follows: resolution of 17,500, AGC of 5e5 and maximum injection time of 100 ms. Other parameters included high collision dissociation energy of 30%, dynamic exclusion for 30 s, as well as single charged, >8 charged and unknown charged ions excluded from fragmentation.

## MS data analysis

Raw files from one biological replicate were searched by one MaxQuant (v 1.5.5.1) session. Database was a reviewed human uniprot database downloaded on 8ᵗʰ May 2018. Carbamidomethylation of cysteine was set as a fixed

modification, while acetylation of protein N-term and oxidation of methionine were set as variable modifications. Peptide and protein false discovery rates were both set at 0.01, and all other parameters were left at default. LFQ (label-free quantification) was enabled but produced many missing values. We then opted for spectral counting for estimation of relative protein abundances (full dataset shown in Supplementary Data 1). The mass spectrometry proteomics data have been deposited to the ProteomeXchange Consortium via the PRIDE partner repository[64], with the dataset identifier PXD038063.

### Immunoprecipitation

Experimentally stimulated cells were lysed in IP lysis buffer containing 1× Complete mini, EDTA-free protease inhibitor (Sigma), 1× PhosSTOP (Sigma), 5 mM N-ethylmaleimide (Sigma), and 10 mM NaF and incubated for 10 min on ice, and the cell lysates were pelleted at $8000 \times g$. Supernatants were incubated with sheep anti-STING overnight at 4 °C. Dynabeads Protein G (Invitrogen) were added to the lysates and incubated for 2 h at 4 °C, followed by three washes with IP lysis buffer containing all the inhibitors. The immunoprecipitated complexes were boiled with 2× Laemmli sample buffer (Sigma) at 95 °C to run immunoblot.

### Cellular fractionation by differential centrifugation

The purification was performed as described previously[11]. Briefly, the cells were collected and washed 3 times in pre-cold 1xPBS. Then, the cell pellets were re-suspended in hypotonic buffer (20 mM HEPES-KOH, pH 7.2, 10 mM KCl, 3 mM MgCl2) plus cocktail protease inhibitors, and the suspensions were homogenized by passing through a 25 G needle. Homogenates were centrifuged at $1000 \times g$ (10 min), and $12,000 \times g$ (10 min). The supernatant was made into 30% Opti-Prep and sample mixture. The discontinuous Opti-Prep gradient was made from bottom to top: 0.5 ml 30% (sample), 0.5 ml 27.5%, 0.5 ml 25%, 1 ml 22%, 1 ml 19%, 0.5 ml 16% and 0.3 ml 12%, and followed by the centrifugation at $150,000 \times g$ for 3 h, using a SW60 Ti Rotor. After centrifugation, each fraction (150 ul) was collected in an orderly fashion from top to bottom. The protein from each fraction was precipitated by methanol-chloroform to be used for the Immunoblotting analysis.

### Luciferase reporter gene assays

THP1-Dual™ cells are derived from the human THP-1 monocyte cell line by stable integration of two inducible reporter constructs (Invivogen). As a result, THP1-Dual™ cells allow the simultaneous study of the NF-κB pathway by monitoring the activity of SEAP and the IRF pathway by assessing the activity of a secreted luciferase (Lucia). Both reporter proteins are readily measurable in the cell culture supernatant when using QUANTI-Blue™ (Invitrogen), a SEAP detection reagent, and QUANTI-Luc™ (Invitrogen), a luciferase detection reagent. In brief, supernatants from THP1-stimulated cells were separately incubated with QUANTI-Blue™ or https://www.invivogen.com/quanti-luc according to the manufacturer's guidelines and read for their luciferase or colorimetric content as a reflection of their IRF and NF-kB activity, respectively.

### LDH cytotoxicity assay

Cell death was measured by lactate dehydrogenase (LDH) release using Pierce LDH Cytotoxicity Assay Kit (ThermoFisher) according to the manufacturer's instructions. Untreated cells were used as a negative control, whereas cells lysed with the provided lysis buffer served as a positive control. LDH activity was measured at 490 nm and 680 nm absorbance using a BioTek Microplate Reader (BioTek Instruments). LDH activity was determined by subtracting the background 680 nm absorbance value from the 490 nm absorbance value. The % of cytotoxicity was calculated using the following formula:

$$\frac{LDH\ activity - LDH\ activity\ negative\ control}{LDH\ activity\ positive\ control - LDH\ activity\ negative\ control} x100$$

### Immunofluorescence of IFIT1 and Cleaved Caspase 3 (separately)

THP1 cells (WT and SAM68 KO) were seeded onto glass coverslips placed on the bottom of a 12-well plate and treated with PMA for further differentiation, as described above. After 24-h incubation, cells were stimulated with cGAMP for 5 h or left untreated for control studies. Afterward, cells were treated for 40 min with 4% PFA following a 20-min permeabilization using 0.2% Triton X-100 in DPBS. Next, blocking with 2% FCS in DPBS was performed for 40 min, and rabbit cleaved caspase 3 antibody (1:400, Cell Signaling) or rabbit IFIT1 antibody (1:800, Cell Signaling) were applied for 1 h at room temperature. After three washes with DPBS, 5 mi each, the cells were incubated with goat anti-rabbit Alexa Fluor 488 nm fluorophore-conjugated secondary antibody (1:400, Invitrogen), Alexa Fluor Plus 647 Phalloidin (1:400, Invitrogen), and PureBlu DAPI Dye (1:100, Bio-Rad) for 1 h at room temperature in the dark. The cells were then washed three times with DPBS and mounted onto microscope slides using ProLong Gold Antifade mountant (Invitrogen). Slides were air-dried in the dark and examined on the next day using a Zeiss LSM 710 Inverted Confocal Microscope with corresponding Zeiss Zen software. Cleaved caspase 3-stained area was measured using ImageJ software.

### Immunofluorescence of SAM68-DKK-FLAG and STING

The cells grown on coverslips were fixed with 4% PFA for 40 min PFA, followed by a 20-minute permeabilization with 0.2% Triton X-100. Next, blocking with 2% FCS in DPBS was performed for 40 min, and afterward samples were incubated with Mouse Monoclonal DDK (FLAG) (OriGene) (1:500 dilution) and Sheep IgG STING/TMEM173 (R&D Systems) (1:50 dilution) antibodies in blocking solution for 1 h at room temperature. After three washes with DPBS, 5 min each, the cells were stained with donkey anti-sheep Alexa Fluor 568 nm fluorophore-conjugated secondary antibody (1:500, Invitrogen), donkey anti-mouse Alexa Fluor 488 nm fluorophore-conjugated secondary antibody (1:500, Invitrogen), and counterstained with PureBlu DAPI Dye (1:100, Bio-Rad) for 1 h at room temperature in the dark. The cells were then washed three times with DPBS and mounted onto microscope slides using ProLong Gold Antifade mountant (Invitrogen). Slides were air-dried in the dark and examined on the next day using a Zeiss LSM 710 Inverted Confocal Microscope with corresponding Zeiss Zen software.

### Immunofluorescence of IRF3 at the mitochondria

The cells grown on glass coverslips were treated with cGAMP for 2 and 5 h or left untreated following fixation with 4% PFA for 40 min and 20-minute permeabilization with 0.2% Triton X-100. Next, blocking solution (2% goat serum (Sigma) in DPBS) was applied for 1 h and afterwards samples were incubated with rabbit anti-IRF3 (Cell signaling) (1:100 dilution) and mouse anti-mitochondria (Abcam) (ab92824) (1:100 dilution) antibodies in blocking solution for 1 h at room temperature. After three washes with DPBS, 5 min each, the cells were stained with goat anti-rabbit Alexa Fluor 488 nm fluorophore-conjugated secondary antibody (1:400, Invitrogen), goat anti-mouse Alexa Fluor 555 nm fluorophore-conjugated secondary antibody (1:500, Invitrogen and counterstained with PureBlu DAPI Dye (1:100, Bio-Rad) for 1 h at room temperature in the dark. The cells were then washed three times with DPBS and mounted onto microscope slides using ProLong Gold Antifade mountant (Invitrogen). Slides were air-dried in the dark and examined on the next day using a Zeiss LSM 800 Inverted Confocal Microscope with corresponding Zeiss Zen software. For IRF3 area quantification within nuclear and mitochondrial compartments, ImageJ software was applied. First, the signal from nuclei or mitochondria was softened (Process → Smooth) to have a homogeneous selection area around it, and then IRF3 area fraction (% area) was quantified within the selection.

### Immunofluorescence of STING and SAM68 at the Golgi

The cells grown on glass coverslips were treated with cGAMP for 2 h or left untreated following fixation with 4% PFA for 40 min and 20-min

permeabilization with 0.2% Triton X-100. Next, blocking solution (2% goat serum (Sigma) in DPBS) was applied for 1 h, and afterwards samples were incubated with rabbit anti-SAM68 (Cell signaling) (1:100 dilution), mouse anti-GM130 (BD biosciences) (1:100 dilution) and sheep IgG STING/TMEM173 (R&D Systems) (1:50 dilution) antibodies in blocking solution for 1 h at room temperature. After three washes with DPBS, 5 min each, the cells were stained with donkey anti-sheep Alexa Fluor 488 nm fluorophore-conjugated secondary antibody (1:500, Invitrogen), donkey anti-rabbit Alexa Fluor 568 nm fluorophore-conjugated secondary antibody (1:500, Invitrogen), donkey anti-mouse Alexa Fluor 647 nm fluorophore-conjugated secondary antibody (1:500, Invitrogen) and counterstained with PureBlu DAPI Dye (1:100, Bio-Rad) for 1 h at room temperature in the dark. The cells were then washed three times with DPBS and mounted onto microscope slides using ProLong Gold Antifade mountant (Invitrogen). Slides were air-dried in the dark and examined on the next day using a Zeiss LSM 800 Inverted Confocal Microscope with corresponding Zeiss Zen software.

## qPCR analysis
Gene expression was analyzed by real-time quantitative PCR. The quality of RNA extracted using the High Pure RNA Isolation Kit (Roche Applied Science) was analyzed by Nanodrop spectrometry using a DS-11 Series Spectrophotometer/Fluorometer (DeNovix). RNA levels for human *IFNB1*, *CXCL10*, *TNFA*, *A20*, *IL6*, and *GAPDH* were analyzed using TaqMan® RNA-to-CT™ 1-Step Kit (Applied Biosystems) according to manufacturer's instructions. GAPD was used as a housekeeping gene.

## KHDRBS1 mRNA synthesis and electroporation delivery
KHDRBS1 in vitro-transcribed (IVT) mRNA was generated by T7 RNA polymerase reaction. The SAM68 human Myc-DKK-tagged ORF (obtained from Origene CAT#: RC200263) was cloned into the IVT plasmid backbone described in[65]. Following plasmid linearization, 3 μl of the digestion reaction was run on a 1% agarose gel to verify successful linearization. The linearized plasmid was precipitated with 5 M Ammonium Acetate and ethanol to concentrate and purify the DNA and subsequently use as a template in the IVT reaction. IVT was performed using the MEGAscript Kit (Ambion, ThermoFisher Scientific) according to the instruction manual, but with full substitution of uridine with pseudouridine (Trilink Biotechnologies or 351 APExBio) and cotranscriptional capping with CleanCap AG (Trilink Biotechnologies) in a 1:4 ratio between GTP and CleanCap. The mRNA was purified and concentrated using the RNA Clean and Concentrator kit (Zymo Research) according to the instruction manual. The quality of the IVT RNA was confirmed on a denaturing formaldehyde gel and quantified by UV-Vis spectrophotometry. 1-2 μg of IVT mRNA were introduced into THP1 cells using a 4D Nucleofection machine (Lonza). P3 Primary Cell Nucleofector Solution and pulsing code CM 138 were used to achieve maximal delivery efficiency.

## KHDRBS1 expressing lentivirus and virus transduction
For the production of lentiviral particles, packaging plasmids pMD2.G, pRSV-Rev, and pMIDg pRRE were transfected along with either pCCL-PGK-KHDRBS1-IRES-Puro or pCCL-PGK-mCherry using standard calcium phosphate transfection in HEK293T cells. Supernatants containing lentiviral particles were harvested and filtered through a 0.45 μm filter. For transduction, $3 \times 10^5$ THP-1 cells were seeded in 12-well plates in 750 μL fresh RPMI 1640 medium. The cells were added 750 μL supernatant containing lentiviral particles expressing KHDRBS1 or mCherry and cultured for 5 days. Expression levels of mCherry as measured by flow cytometry were used as a pseudo-indicator of transduction efficiency and were generally above 80% mCherry positive cells.

## Statistics and reproducibility
Graphs were generated using GraphPad Prism 9 (GraphPad Software). The data are shown as means of biological replicates ± s.e.m. The data are the means of 2–4 independent experiments performed in 3–6 biological replicates unless otherwise mentioned in the figure legends. If indicated, data were analyzed for statistically significant differences between groups using a two-tailed unpaired Student's $t$ test (**$p < 0.01$; ***$p < 0.001$; ****$p < 0.0001$) or a two-tailed one-way ANOVA followed by Sidak's multiple comparison test (**$p < 0.01$; ***$p < 0.001$; ****$p < 0.0001$).

## Reporting summary
Further information on research design is available in the Nature Portfolio Reporting Summary linked to this article.

## Data availability
The mass spectrometry proteomics data have been deposited to the ProteomeXchange Consortium via the PRIDE partner repository[64], with the dataset identifier PXD038063. The data can also be seen in Supplementary Data 1. All source data underlying the graphs in the main figures can be found in Supplementary Data 2. The graphical abstract and Fig. 3k were created using BioRender.com. All other data that support the findings of this study are available from the corresponding authors upon reasonable request.

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

## Acknowledgements

The work was supported by grants from The European Research Council (786602); Independent Research Fund Denmark (1026-00003B); The Novo Nordisk Foundation (NNF18OC0030274) to S.R.P.; The Lundbeck Foundation (R335-2019-2138), the Kræftens Bekæmpelse (R279-A16218), the Brødrene Hartman Fonden, the Hørslev Fonden, the Einar Willumsens mindelegat, the Eva og Henry Frænkels mindefond, the Beckett Fonden, the Lily Benthine Lunds fonden, the Sofus Carl Emil Friis Legat to D.O. The PhD scholarship to D.v.d.H. was partly funded by the Department of Biomedicine, Aarhus University. D.v.d.H. was also supported by the Dansk Kræftforskningsfond. The mass spectrometry work was funded by the Novo Nordisk Foundation (NNF18OC0032724) and the Carlsberg Foundation to R.A.F. The assistance of Ea Stoltze Andersen and Kirsten Stadel Petersen is greatly appreciated. The authors also would like to thank the FACS core facility at Aarhus University, Department of Biomedicine for its technical support.

## Author contributions

D.O. and S.R.P. conceived the idea and designed the experiments. D.v.d.H., N.K., M.H.S.M., S.A., Q.W., J.Z., E.C., A.H.F.R, M.B.I., F.R., T.I.J., R.N., L.S.d.C. C.F.K., V.R.S. and B-C.Z. performed the experiments. R.O.B. C.S., R.A.F., J.G.M, D.O., and S.R.P. supervised experiments. D.O. and S.R.P. interpreted data with input from all authors. D.O. and S.R.P. wrote the draft of the manuscript, and all authors contributed to the final version.

## Competing interests

The authors declare no competing interests.

## Ethics approval

All collaborators of this study have fulfilled the criteria for authorship required by Nature Portfolio journals have been included as authors, as their participation was essential for the design and implementation of the study. Roles and responsibilities were agreed among collaborators ahead of the research. This work includes findings that are locally relevant, which have been determined in collaboration with local partners. This research was not severely restricted or prohibited in the setting of the researchers, and does not result in stigmatization, incrimination, discrimination or personal risk to participants. Local and regional research relevant to our study was taken into account in citations.
