## [Peer Review File · Communications Biology]

Reviewers' comments:

Reviewer #1 (Remarks to the Author):

This manuscript by van der Horst et al., identifies SAM68 as a new factor controlling STING-induced cell death in macrophages. The authors examine numerous unbiased screens including their own to identify SAM68 as a factor mediating intrinsic (mitochondrial) apoptosis in response to STING. While not required for canonical STING immune responses, SAM68 is involved in inducing apoptosis. Unexpectedly, SAM68 does not induce cell death via one of its known functions but instead appears to directly interact with STING to link TBK1/IRF3 to apoptosis. This interesting manuscript adds to a growing body of work in the area of cell death induced downstream of STING activation, which appears to have multiple mechanisms across different cell types. This is a logical and generally well controlled study; however, some important additional experiments are required. In addition, given this manuscript makes some contrasting observations to previously published studies, these discrepancies need to be clearly addressed in the discussion i.e., what is different in study X and suggestions as to why. Finally, as the mechanisms of STING induced cell death are varied in cell types and remain unclear, a little more experimental examination of precisely how STING-SAM68-TBK1-IRF3 = apoptosis would be valuable.

Major points:

1. Figure 1H: Examination of cell viability in BAX/BAK dKO cells would give a definitive answer as to if this is indeed intrinsic apoptosis and should be undertaken.
2. Figure 2: The dependence of IRF3 activity but not IFNAR signaling or translation fits with previous findings from MAVS activation, where IRF3 can activate BAX to elicit apoptosis (PMID: 20360684). Indeed, the authors state "(line 144) One of the first studies to identify STING-dependent apoptosis reported formation of a complex between BAX and IRF3 in primary human monocytes, inducing apoptosis through the intrinsic pathway (Sze et al., 2013)". I recommend this be examined here as part of the mechanism or at least discussed. E.g., does any IRF3 relocate to the OMM upon STING activation?
3. Figure 2I: The reconstitution of SAM68 in KO cells with in vitro transcribed mRNA appears to only lead to a small amount of SAM68 produced, although enough to rescue casp-3 cleavage to WT levels etc. Why is the band corresponding to SAM68 higher than the WT band? Please explain.
4. Figure 2J-L: Some previous work in BMDMs has shown these cells do not die upon STING activation (PMID: 28874664). Is this different in PMs? The authors should also examine the effects of STING activation on BMDM viability. Do BMDMs also express SAM68? Why the difference? This is an important finding to make as many in the field rely on BMDMs as a representative for tissue resident murine macrophages.
5. Figure 2K: The blots are missing labels making it hard to interpret these data. Please fix.
6. Figure 4E: Imaging SAM68 with GM130 & STING would greatly strengthen this finding.

Minor points:

7. Introduction, line 75: The ref PMID: 32268090 should be added that extends the mechanism of STING-NFkB. Also, the authors could mention the seminal studies (PMID: 32640258, PMID: 32636381) that identify critical non-IFN roles of STING.
8. Introduction, line 80: The findings of two new studies demonstrating ESCRT-dependent STING degradation could also be added here (PMID: 36918692, PMID: 36739287).
9. Discussion: The findings from figure 1 are in contrast to previous work in human monocytes/macrophages (PMID: 29033128). This discrepancy should be discussed.
10. Results, line 127: Should also cite work on cell death in DCs (PMID: 35281062).
11. SAM68 appears to only be partially responsible for mediating STING-induced activation – this should be made clear throughout the manuscript.

Reviewer #2 (Remarks to the Author):

In this manuscript, the authors explore the mechanisms by which STING induces apoptosis in macrophages. They report that STING interacts with SAM68 to induce IRF-3 dependent mitochondrial apoptosis pathway in an independent-manner of the protein de novo synthesis pathway. Their data suggests that this interaction occurs in the Golgi and the initiation of STING-induced apoptosis happens after exiting the Golgi. In addition, they can conclude that these effects are independent of the IFN and cytokines signaling. Altogether, the authors show that there is a link between SAM68 and STING activation that leads to the induction of apoptosis through an intrinsic pathway.

Comments and questions:

- The authors state from Figure 1 onward that STING induces cell death through the mitochondrial apoptosis pathway. To explore this, the authors use LDH release (%), flow cytometry and proteins involved in the apoptotic cascade by western blot in the presence or absence of cGAMP or dsDNA. Even though there is extensive literature of the effects of cGAMP or dsDNA on STING signaling, the authors should provide in the western blot panels the expression levels of STING to demonstrate the validation of a successful treatment as well as demonstration of the modulation of STING (Figure 1 panels C and J). Similarly, they should include the expression levels of STING in the western blot panels of Figure 2 (panels C, D and I) and Figure 3 panel C.
- The authors identified 7 proteins that could be involved in STING induced apoptosis. To explore the effects of these proteins in apoptosis, they treated the cells with two specific gRNAs that impaired these 7 genes. However, the authors do not provide any validation in their readouts of the impairment of most of these targets (except for SAM68). A supplementary panel on figure S2 showing the validations of these cell lines either at the genomic level or by western blot expression of the target should be included.
- Please provide the labels/legend for the gray and green bar graphs as they are missing from the graph in Figure 1 panel H.
- Please provide a western blot of IRF3 expression levels on Figure 1 panel K.
- The protein names of each blot are missing in Figure 2 panel K.
- Please explain why the levels of IRF3 in the STING immunoprecipitation (Figure 4 panel A) are not sustained after cGAMP treatment even though STING, TBK1 and SAM68 have a sustained induction over a 4h period.

Reviewer #1 (Remarks to the Author):

This manuscript by van der Horst et al., identifies SAM68 as a new factor controlling STING-induced cell death in macrophages. The authors examine numerous unbiased screens including their own to identify SAM68 as a factor mediating intrinsic (mitochondrial) apoptosis in response to STING. While not required for canonical STING immune responses, SAM68 is involved in inducing apoptosis. Unexpectedly, SAM68 does not induce cell death via one of its known functions but instead appears to directly interact with STING to link TBK1/IRF3 to apoptosis. This interesting manuscript adds to a growing body of work in the area of cell death induced downstream of STING activation, which appears to have multiple mechanisms across different cell types. This is a logical and generally well controlled study; however, some important additional experiments are required. In addition, given this manuscript makes some contrasting observations to previously published studies, these discrepancies need to be clearly addressed in the discussion i.e., what is different in study X and suggestions as to why. Finally, as the mechanisms of STING induced cell death are varied in cell types and remain unclear, a little more experimental examination of precisely how STING-SAM68-TBK1-IRF3 = apoptosis would be valuable.

We would like to thank Reviewer 1 for the positive nature of the comments raised on our manuscript. We fully agree that our manuscript does make some contrasting observations to the already published literature on STING-mediated cell death. We have now highlighted the similarities/discrepancies of our findings with the existing literature in the revised version of our discussion.

Major points:

1. Examination of cell viability in BAX/BAK dKO cells would give a definitive answer as to if this is indeed intrinsic apoptosis and should be undertaken.

To unequivocally confirm that STING activation led to intrinsic apoptosis in cGAMP-stimulated PMA-differentiated THP1, cells were double Knock-Out (dKO) for BAX and BAK1 using a CRISPR/Cas9 gene editing strategy. dKO of BAX and BAK1 led to a complete abrogation of LDH release and a reduction in cleavage of Casp3 and PARP in cGAMP-stimulated PMA-differentiated THP1 cells, thus confirming the involvement of the intrinsic pathway in triggering STING-induced apoptosis. The results from this experiment are now displayed in the revised Fig. 1k-l of the manuscript.

2. The dependence of IRF3 activity but not IFNAR signaling or translation fits with previous findings from MAVS activation, where IRF3 can activate BAX to elicit apoptosis (PMID: 20360684). Indeed, the authors state “(line 144) One of the first studies to identify STING-dependent apoptosis reported formation of a complex between BAX and IRF3 in primary human monocytes, inducing apoptosis through the intrinsic pathway (Sze et al., 2013)”. I recommend this be examined here as part of the mechanism or at least discussed. E.g., does any IRF3 relocate to the OMM upon STING activation?

We fully agree with the point of the reviewer. We have now included the citation from Chattopadhyay *et al*, 2010 in our introduction and we have expanded our discussion section

on IRF3 relocation to the mitochondria and IRF3-BAX interaction in apoptosis induction following STING engagement (lines 111-113 and lines 325-326).

In addition, and to gain more insights into the mode of action of STING-mediated apoptosis in PMA-THP1 cells, we investigated the relocation of IRF3 to the mitochondria following cGAMP stimulation at different time points by confocal microscopy. The data of this experiment is now included in the revised manuscript Fig. 2h and i.

Finally, in contrast to previous reports (Chattopadhyay et al, 2010, EMBO J; Sze et al, 2013, Cell Host and Microbe), we could not find a potent induction of IRF3-Bax interaction following cGAMP stimulation in THP1 cells, however, we did observe that the strength of the interaction was to some extent SAM68 dependent and that some level of IRF3-Bax interaction existed at a resting state in PMA-THP1 cells. The data are appended below.

3. The reconstitution of SAM68 in KO cells with *in vitro* transcribed mRNA appears to only lead to a small amount of SAM68 produced, although enough to rescue casp-3 cleavage to WT levels etc. Why is the band corresponding to SAM68 higher than the WT band? Please explain.

SAM68 mRNA was *in vitro* transcribed. To do so, a plasmid expression vector from Origene (CAT#RC200263) was cloned into an *in vitro* transcription plasmid (IVT) construct as reported in the methodology section. The original SAM68 plasmid expression vector carries a Myc-DKK tag which explains the higher molecular weight of reconstituted SAM68 when compared to the endogenous protein. The reference of the original plasmid carrying the Myc-DKK tag has been added to the methodology section.

Additionally, the reviewer mentioned that “the reconstitution of SAM68 in KO cells with *in vitro* transcribed mRNA appears to only lead to a small amount of SAM68 produced, although enough to rescue caspase 3 cleavage to WT levels”. To strengthen and consolidate our SAM68 reconstitution findings, we used another approach to express SAM68 in KO cells, this time using a lentiviral vector as a delivery platform. The data are displayed in the revised Fig. 3j and demonstrate that the restoration of SAM68 protein expression increases the levels of cleaved caspase 3 and PARP proteins in response to cGAMP stimulation in SAM68 KO THP1 cells.

4. Some previous work in BMDMs has shown these cells do not die upon STING activation (PMID: 28874664). Is this different in PMs? The authors should also examine the effects of STING activation on BMDM viability. Do BMDMs also express SAM68? Why the difference? This is an important finding to make as many in the field rely on BMDMs as a representative for tissue resident murine macrophages.

The reviewer is indeed correct that Gulen *et al* reported that BMDM did not die following 16-hour stimulation with the mouse-specific STING agonist CMA. In our work, the set-up is slightly different since animals were injected intraperitoneally with cGAMP prior to peritoneal macrophages (PMs) collection and assessment of cleavage of caspase 3 by immunoblotting. This is an important point to raise since the experiments are clearly not directly comparable here.

However, to address the point from the reviewer, we differentiated bone marrow (BM) cells into bone marrow-derived macrophages (BMDMs) and tested for cleavage of caspase 3 by immunoblotting and for cell death by LDH release assay following cGAMP stimulation. We also measured the levels of SAM68 by immunoblotting in BMDMs, which we found to be highly expressed. The data show that under our experimental conditions, cGAMP was found to induce cell death in BMDMs, with increased LDH release and levels of the apoptotic marker cleaved Caspase 3. The data are displayed in a new Figure S3 and the extended discussion is displayed on lines 285-290.

5. The blots are missing labels making it hard to interpret these data. Please fix.

The labels (cleaved-CASP3 and STING) have now been added to Figure 2L in the revised version of the figures.

6. Imaging SAM68 with GM130 & STING would greatly strengthen this finding.

To gain more insights into the mode of action of STING/SAM68-mediated apoptosis, we completed a series of experiments looking at the co-localization of STING and SAM68 in the Golgi compartment following cGAMP stimulation by confocal microscopy. The data of these experiments are now included in the revised Fig. 5.

Minor points:

7. Introduction, line 75: The ref PMID: 32268090 should be added that extends the mechanism of STING-NFkB. Also, the authors could mention the seminal studies (PMID: 32640258, PMID: 32636381) that identify critical non-IFN roles of STING.

The different suggested references have been added to the revised version of the introduction.

8. Introduction, line 80: The findings of two new studies demonstrating ESCRT-dependent STING degradation could also be added here (PMID: 36918692, PMID: 36739287).

The findings of the two studies demonstrating ESCRT-dependent STING degradation by microautophagy have now been properly introduced in the revised version of our introduction.

9. Discussion: The findings from figure 1 are in contrast to previous work in human monocytes/macrophages (PMID: 29033128). This discrepancy should be discussed.

We added a section where we are now discussing the similarities and discrepancies between both our study and the study of Gaidt *et al.* (lines 296-311). However, we would like to point towards a few things that complicate the comparison of these two studies.

First of all, our findings were raised from experiments in PMA-differentiated THP1 cells, primary human monocyte-derived macrophages, and primary murine macrophages or BMDMs, while the vast majority of the experiments from Gaidt *et al.* were performed on BLaER1 cells, a malignant B-cell line that is trans-differentiated into a monocytic like cell line. Additionally, our manuscript predominantly used LDH release, cleavage of caspase 3/PARP and Annexin V/PI staining as readouts to draw our conclusions while the work by Gaidt *et al.* used the release of the pro-inflammatory cytokine IL1b instead.

Overall, while Z-VAD was not able to suppress STING-induced cell death in BLaER1 cells in the manuscript by Gaidt *et al.*, only the pan-caspase inhibitor Z-VAD and not Caspase 1, NLRP3, or necroptosis inhibitors could prevent STING-induced cell death as measured by LDH release in PMA-THP1 and primary human MDMs models in our study. Thus, STING activation can induce several modalities of cell death including apoptosis, lysosomal cell death, autophagic cell death, necroptosis, and pyroptosis but there seems to be a significant cell-type specificity regarding the modality and mechanism of STING-induced cell death which could explain some of the differences observed in both manuscripts.

10. Results, line 127: Should also cite work on cell death in DCs (PMID: 35281062).

The work from Pang *et al.*, 2022 has been cited in the revised version of our results section.

11. SAM68 appears to only be partially responsible for mediating STING-induced activation – this should be made clear throughout the manuscript.

We have altered our revised text to make the point clear that SAM68 is only partially involved in mediating STING-induced apoptosis.

Reviewer #2 (Remarks to the Author):

In this manuscript, the authors explore the mechanisms by which STING induces apoptosis in macrophages. They report that STING interacts with SAM68 to induce IRF-3 dependent mitochondrial apoptosis pathway in an independent-manner of the protein de novo synthesis pathway. Their data suggests that this interaction occurs in the Golgi and the initiation of STING-induced apoptosis happens after exiting the Golgi. In addition, they can conclude that these effects are independent of the IFN and cytokines signaling. Altogether, the authors show that there is a link between SAM68 and STING activation that leads to the induction of apoptosis through an intrinsic pathway.

Comments and questions:

- The authors state from Figure 1 onward that STING induces cell death through the mitochondrial apoptosis pathway. To explore this, the authors use LDH release (%), flow cytometry and proteins involved in the apoptotic cascade by western blot in the presence or absence of cGAMP or dsDNA. Even though there is extensive literature of the effects of cGAMP or dsDNA on STING signaling, the authors should provide in the western blot panels the expression levels of STING to demonstrate the validation of a successful treatment as well as demonstration of the modulation of STING (Figure 1 panels C and J). Similarly, they should include the expression levels of STING in the western blot panels of Figure 2 (panels C, D and I) and Figure 3 panel C.

We agree with this point from Reviewer 2 and we have now tried to accommodate our manuscript in the best way possible. For this reason, we have updated all panels in the manuscript, which are now displaying markers for both STING-mediated immune activation (including p-STING/STING, p-TBK1/TBK1, p-IRF3/IRF3, and ISG15 levels) and STING-induced apoptosis (CL-CASP3 and CL-PARP). In order to maintain a logical flow for our manuscript, we felt that Figure 3 should only focus on apoptosis in SAM68 wt vs KO cells. However, we are appending below the full WB panels for experiments 3c and 3d confirming that STING pathway activation led to immune activation following dsDNA or cGAMP stimulation.

- The authors identified 7 proteins that could be involved in STING induced apoptosis. To explore the effects of these proteins in apoptosis, they treated the cells with two specific gRNAs that impaired these 7 genes. However, the authors do not provide any validation in their readouts of the impairment of most of these targets (except for SAM68). A supplementary panel on figure S2 showing the validations of these cell lines either at the genomic level or by western blot expression of the target should be included.

This point is highly valid. Therefore, we are pleased to now provide all the validation data of KO efficiency at the genomic level for the different guides RNA tested. We performed a ICE analysis and at least one guide out of the two tested satisfied an indel rate and a KO rate of 50% or higher for most of the genes tested. These validation data are now included in the revised Figure S2 of the manuscript.

- Please provide the labels/legend for the gray and green bar graphs as they are missing from the graph in Figure 1 panel H.

The figure 1H is not part of the manuscript anymore and has been replaced by an experiment where we performed double KO of BAX and BAK1 presented in Figure 1.

- Please provide a western blot of IRF3 expression levels on Figure 1 panel K.

An immunoblot of IRF3 expression levels has been added to the revised Fig. 2G. On top of that, we also expanded on the immunologic markers for this panel and included P-STING/STING, P-TBK1/TBK1, P-IRF3/IRF3, and ISG15.

- The protein names of each blot are missing in Figure 2 panel K.

The labels (cleaved-CASP3 and STING) have now been added to the revised version of the figure.

- Please explain why the levels of IRF3 in the STING immunoprecipitation (Figure 4 panel A) are not sustained after cGAMP treatment even though STING, TBK1 and SAM68 have a sustained induction over a 4h period.

The authors have also noted that the association of IRF3 with STING exhibits different kinetics than TBK1 and SAM68. Although this cannot be unequivocally explained with the available data, it is likely that IRF3 dissociates from STING upon phosphorylation at S386 and forms homodimers and translocates to the nucleus (PMID: 33205822). This is not the case for TBK1 (and likely also not for SAM68). On the other hand, it is important to note that the levels of total STING are also decreased 4 h post-stimulation, which means that on a per-molecule-basis, a higher proportion of STING molecules are associated with IRF3 at 4 h compared to control.

Reviewers' comments:

Reviewer #1 (Remarks to the Author):

While the authors have made significant efforts to address my original concerns there are some major issues with the new data and thus the conclusions drawn. I am not convinced that the data fully supports the overall findings and the appropriate scientific rigor is missing in many of the new (and original) experiments. I think this manuscript requires major revisions before reconsideration. See below:

Major point 1 (original revision): I greatly appreciate the authors attempting to generate Bax/Bak dKO THP-1 cells to address the notion that STING-induced death is indeed via intrinsic apoptosis. However, there are a few major issues with the new data presented. Firstly, although the LDH data (Fig.1k) supports the hypothesis, the WB data does not (Fig.1j). Hence, I disagree with the conclusions drawn by the authors.

- By WB, the BAK1 levels do not appear to be reduced compared to WT, BAK1 KO or B/B dKO – how did you validate its KO?
- Caspase-3 and PARP are downstream of B/B, however by WB in the dKO cells, they are both still cleaved, albeit reduced (but same as in BAK1 KOs) – i.e., they appear functional.
- Also, and perhaps most importantly, there is no positive control for intrinsic apoptosis (e.g., ABT-373 + Mcl1 inhibitor) to demonstrate the validity of the B/B dKOs. This should be included in both experiments given the unclear nature of the WB data.
- LDH data is from 2 experiments? So what do the 12 dots represent? And how can stats be performed on only 2 independent experiments? WB data from n=1 only. No conclusions should be drawn from n=1. This should be repeated at least so n=3 with controls!

Major point 2 (original revision): The new data in Fig.2h-i is not convincing and lacks rigor (only n=1). This should be repeated at least so n=3! This is true of all experiments in the manuscript – should be n=3.

- What is the exact marker used to examine the mitochondria? Is it OMM? Method states “mouse anti-mitochondria (Abcam)”?
- The representative image shows 1 of ~20 cells having IRF3/Mito co-localisation so its hard to say that this is a real event. Also, single channel images of the ROI would be helpful.
- The quantification of data is unusual and not very informative. It is stated in the Fig legend “The heatmaps display the values for each individual cell.” Co-localisation of the channels should be assessed over 3 individual experiments and then combined with SEM.
- An additional and clearer approach may be to isolate mitos after +/- STING activation and look for IRF3 by WB.

Major point 4 (original revision): I appreciate the authors attempting to address my Q regarding cell death in BMDMs. However, this raised some questions about the in vitro data in the manuscript.

- I noted that in generating the new Figure S3 in which modest cell death is observed (low LDH, low CL-C3) in BMDMs after STING activation, the authors used 100 µg/mL of cGAMP added to the extracellular space. This is extremely high, as typically, 5-10 µg/mL is used to activate STING i.e., here 10-20x more than usual was used. Why so much? Was this to avoid transfection? If so, it might be more advantageous and experimentally consistent to activate with a modified cGAMP (e.g., 2'3'-cGAM(PS)₂ that is stable in the extracellular space) and also diABZI (or other synthetic ligand) at a lower dose. A dose response of STING ligand here and in THP-1 should be performed to show when death is induced. Is it only at very high concentrations? Is it physiological?

Major point 6 (original revision): I am not convinced by new Fig.5f. Under basal conditions, there is not a strong correlation between GM130 and SAM68, but this looks the same even following activation with cGAMP – i.e., no SAM68 in Golgi? 2h cGAMP should also result in STING/GM130 co-localisation but this does not appear to be the case. The DNA stain is distracting in the merged images. Also, n=2

only!

Reviewer #2 (Remarks to the Author):

In this manuscript, the authors report that STING interacts with SAM68 to induce IRF-3 dependent mitochondrial apoptosis pathway in an independent-manner of the protein de novo synthesis pathway. Their data suggests that this interaction and the initiation of STING-induced apoptosis happens after exiting the Golgi. Altogether, the authors show that there is a link between SAM68 and STING activation that leads to the induction of apoptosis through an intrinsic pathway.

The authors have addressed all the comments and questions by the reviewers. The new experiments, the edited discussion and the references incorporated have significantly improved the article. No further questions or comments needs to be addressed.

Reviewers' comments:

Reviewer #1 (Remarks to the Author):

While the authors have made significant efforts to address my original concerns there are some major issues with the new data and thus the conclusions drawn. I am not convinced that the data fully supports the overall findings and the appropriate scientific rigor is missing in many of the new (and original) experiments. I think this manuscript requires major revisions before reconsideration. See below:

We were delighted to read that the reviewer appreciated the efforts made to satisfy his/her original concerns. However, there seemed to be some additional experimental requests that we were happy to address in the point-by-point response below.

Major point 1 (original revision): I greatly appreciate the authors attempting to generate Bax/Bak dKO THP-1 cells to address the notion that STING-induced death is indeed via intrinsic apoptosis. However, there are a few major issues with the new data presented. Firstly, although the LDH data (Fig.1k) supports the hypothesis, the WB data does not (Fig.1j). Hence, I disagree with the conclusions drawn by the authors.

- By WB, the BAK1 levels do not appear to be reduced compared to WT, BAK1 KO or B/B dKO – how did you validate its KO?

We agree with the reviewer's comment that in our initial set of data (Fig. 1l) the protein levels of BAK1 were only moderately altered in the CRISPR/Cas9-edited cells compared to the levels in the WT cells. We have performed a new set of experiments where we have tried to optimize the delivery and the KD efficiency of BAX/BAK1 protein in THP1 cells. We also used cells that were electroporated with Cas9 and an irrelevant sequence (AAVS1) as control (see data below). Although the disruption of BAX was almost complete, there is still some residual BAK1 protein with the methodology used.

New data assessing cGAMP-induced release of LDH and cleavage of Caspase 3 and PARP in pools of cells subjected to BAX/BAK1 genome-editing *versus* AAVS1 controls is now reported in Fig.1k and 1l.

- Caspase-3 and PARP are downstream of B/B, however by WB in the dKO cells, they are both still cleaved, albeit reduced (but same as in BAK1 KO) – i.e., they appear functional.

We fully agree with this point from the reviewer that cleavage of PARP and CASP3 was only moderately attenuated and not completely abolished in the double KO cells compared to the WT cells following cGAMP treatment in our initial collection of data.

New data with improved delivery of Cas9 and gRNA by electroporation and the use of AAVS1-treated cells as control cells lead to clear reduction of Caspase 3 and PARP cleavage. The new data are displayed in Fig. 1k. Although the effect is still not complete, the cellular phenotype is consistent with the rest of the data presented, and support the conclusions drawn.

One explanation why we do not observe complete loss of Caspase 3/PARP cleavage could be that we are not working with KO cell clones but rather with a bulk population of cells treated with gRNAs targeting two genes. Not all cells from the bulk are fully KO for both proteins and this could explain some of the residual activity seen in terms of Caspase3 and PARP cleavage. Alternatively, another pathway than the BAX/BAK mitochondrial pathway could also be at play in cGAMP-treated cells leading to the observed residual cleavage of Caspase3 and PARP.

- Also, and perhaps most importantly, there is no positive control for intrinsic apoptosis (e.g., ABT-373 + Mcl1 inhibitor) to demonstrate the validity of the B/B dKOs. This should be included in both experiments given the unclear nature of the WB data.

As suggested by the reviewer, we have now included the positive control for intrinsic apoptosis (ABT-737 + S-63845) in our experiments. The new data are displayed in Fig. S2f-g.

- LDH data is from 2 experiments? So what do the 12 dots represent? And how can stats be performed on only 2 independent experiments? WB data from n=1 only. No conclusions should be drawn from n=1. This should be repeated at least so n=3 with controls!

The original LDH data were from two independent experiments each performed in 6 biological replicates (6 individual wells per experiment). The WB data were from one representative experiment that had been repeated twice (one from each individual experiment of the LDH). We now have repeated independently 4 experiments performed in multiple biological replicates as suggested by the reviewer. The data are shown in Fig. 1l.

Major point 2 (original revision): The new data in Fig.2h-i is not convincing and lacks rigor (only n=1). This should be repeated at least so n=3! This is true of all experiments in the manuscript – should be n=3.

We repeated experiment 2h-i so that n=3 in the revised version of the manuscript. Quantification is also from three independent experiments and represents the means +/- SEM. New data are shown in Fig. 2h-i.

- What is the exact marker used to examine the mitochondria? Is it OMM? Method states “mouse anti-mitochondria (Abcam)”?

For this confocal experiment, we have used the Anti-Mitochondria antibody (113-1) – BSA and Azide free (ab92824) from Abcam as appropriately referenced in our manuscript. We have contacted Abcam directly and the response below has been obtained from Korinn Murphy, PhD, a Scientific Support Specialist in the company. According to her words “we list the immunogen for our antibody as human cell homogenate. The exact target detected by the antibody is not known. Since human mitochondrial extract was used as the antigen, we therefore suspect that the target is not a single protein, but rather likely key multiple proteins that are specific to mitochondria”. As can be seen from the vendor’s webpage, this antibody has been referenced 77 times in peer-reviewed publications.

- The representative image shows 1 of ~20 cells having IRF3/Mito co-localisation so its hard to say that this is a real event. Also, single channel images of the ROI would be helpful.

We do not fully agree with this comment from the reviewer. We did indeed magnify one of the cells to show the overlap between IRF3 and the mitochondrial staining. However, there was a greater number of cells in this field of about 30 cells that displayed some overlap between IRF3 and the mitochondrial staining. This observation also aligned with our quantification (cf point below). From our quantification, about 5-7 cells/25 cells (about 20–30%) displayed overlap between staining for IRF3 and mitochondria 5h post cGAMP stimulation. This aligned quite reasonably with the percentage of dead/apoptotic cells after 18-24h cGAMP stimulation in THP1 cells.

We have completed a new series of experiment at 3 and 6 hrs following stimulation with cGAMP. We are also providing the ROI in the revised version of the manuscript as suggested by the reviewer. (Fig. 2)

- The quantification of data is unusual and not very informative. It is stated in the Fig legend “The heatmaps display the values for each individual cell.” Co-localisation of the channels should be assessed over 3 individual experiments and then combined with SEM.
- An additional and clearer approach may be to isolate mitos after -/+ STING activation and look for IRF3 by WB.

We agree that this quantification was not the standard way of showing this type of data. However, we honestly found that this way of quantifying the data was more informative than a more traditional “overlay way”. Our choice to represent the quantification of IRF3 at the nucleus/mitochondria for individual cells rather than representing the general co-localisation values of the entire channels was motivated by the fact that not all cells stimulated with cGAMP in the population become apoptotic or positive for IRF3 at the mitochondria. Thus, our representation gave a more accurate overview of what happened at the single-cell level in the population of cGAMP-stimulated cells. As to why only 20-30% cells become apoptotic when more than 80% are IRF3+ in the nucleus and have successfully received cGAMP, this remained unexplained and could warrant future attention of researchers in follow-up studies. In accordance with the reviewer’s suggestion, we performed and displayed co-localization quantification of the channels studied. Data are presented in Fig. 2i.

Major point 4 (original revision): I appreciate the authors attempting to address my Q regarding cell death in BMDMs. However, this raised some questions about the in vitro data in the manuscript.

- I noted that in generating the new Figure S3 in which modest cell death is observed (low LDH, low CL-C3) in BMDMs after STING activation, the authors used 100 µg/mL of cGAMP added to the extracellular space. This is extremely high, as typically, 5-10 µg/mL is used to activate STING i.e., here 10-20x more than usual was used. Why so much? Was this to avoid transfection? If so, it might be more advantageous and experimentally consistent to activate with a modified cGAMP (e.g., 2'3'-cGAM(PS)₂ that is stable in the extracellular space) and also diABZI (or other synthetic ligand) at a lower dose. A dose response of STING ligand here and in THP-1 should be performed to show when death is induced. Is it only at very high concentrations? Is it physiological?

This is an excellent point from the reviewer. We did not in any way try to avoid transfection in our experimental model. The choice of exogenous treatment by cGAMP in our experiments was motivated by the simplicity of the experiment where the molecule could be added straight to the culture medium and also by the costs saved from not having to use an expensive transfection reagent. However, we have completed a series of new experiments showing that 100 µg/mL of exogenous cGAMP treatment roughly aligns with a transfection of cGAMP at 10 µg/mL (using lipofectamine) in terms of immune response activation (phosphorylation of STING and IRF3) and induction of apoptosis (cleavage of caspase 3 and PARP) both in THP1 cells and BMDMs (See Fig. S1a and S5c).

We also performed the dose-response experiments suggested by the reviewer in both THP1 and BMDMs using an exogenous treatment of cGAMP, cGAMP(PS)₂ and diABZI. The pattern of cell death induction is similar for all three drugs although the modified cGAMP and diABZI seem to act at lower concentrations and also probably faster (in light of STING degradation/phosphorylation). The new data are now included in Fig. S1b and S5d.

Major point 6 (original revision): I am not convinced by new Fig.5f. Under basal conditions, there is not a strong correlation between GM130 and SAM68, but this looks the same even following activation with cGAMP – i.e., no SAM68 in Golgi? 2h cGAMP should also result in STING/GM130 co-localisation but this does not appear to be the case. The DNA stain is distracting in the merged images. Also, n=2 only!

We now repeated the experiment three times with a similar trend. Additionally, the DNA stain is now excluded from the merged images. Of note, the interaction between SAM68 and STING is probably very short hence explaining the difficulty to catch it up by confocal microscopy. More advanced technologies such as ImageStream could in the future help capturing this interaction better. However, in light of the different readouts used in Figure 5 co-IP, confocal microscopy and fractionation experiments, we are quite convinced that there is some level of interaction between STING and SAM68 happening post trafficking of STING at the Golgi compartment.

Reviewer #2 (Remarks to the Author):

In this manuscript, the authors report that STING interacts with SAM68 to induce IRF-3 dependent mitochondrial apoptosis pathway in an independent-manner of the protein de novo synthesis pathway. Their data suggests that this interaction and the initiation of STING-induced apoptosis happens after exiting the Golgi. Altogether, the authors show that there is a link between SAM68 and STING activation that leads to the induction of apoptosis through an intrinsic pathway.

The authors have addressed all the comments and questions by the reviewers. The new experiments, the edited discussion and the references incorporated have significantly improved the article. No further questions or comments needs to be addressed.

We are thankful to Reviewer 2 who is satisfied with the revisions applied to the manuscript.

REVIEWERS' COMMENTS:

Reviewer #1 (Remarks to the Author):

I thank the authors for all their hard work in addressing my concerns. For the most part I am satisfied that they have done what is required to warrant publication.

I would however recommend the authors acknowledge the partial KD of B/B in Fig.1k and S2f-g and perhaps mention why this is may be the case in the text or discussion (as was stated in the letter).

REVIEWERS' COMMENTS:

Reviewer #1 (Remarks to the Author):

I thank the authors for all their hard work in addressing my concerns. For the most part I am satisfied that they have done what is required to warrant publication.

I would however recommend the authors to acknowledge the partial KD of B/B in Fig.1k and S2f-g and perhaps mention why this is may be the case in the text or discussion (as was stated in the letter).

We are thankful to Reviewer 1 who is satisfied with the revisions applied to the manuscript. As suggested by the reviewer, we acknowledged the partial KD of B/B in Fig. 1k and mentioned the different possibilities explaining this observation in the revised and final version of the discussion.